# Predict the Phase Angle Master Curve and Study the Viscoelastic Properties of Warm Mix Crumb Rubber-Modified Asphalt Mixture

**DOI:** 10.3390/ma13215051

**Published:** 2020-11-09

**Authors:** Fei Zhang, Lan Wang, Chao Li, Yongming Xing

**Affiliations:** 1School of Science, Inner Mongolia University of Technology, Hohhot 010040, China; 20171000002@imut.edu.cn; 2Key Laboratory of Civil Engineering Structure and Mechanics, Inner Mongolia University of Technology, Hohhot 010051, China; wl8802616@aliyun.com (L.W.); nmggydx@21cn.com (C.L.); 3School of Civil Engineering, Inner Mongolia University of Technology, Hohhot 010040, China

**Keywords:** dynamic modulus, phase angle, shift factor, master curve, Kramers–Kronig relations

## Abstract

To identify the most accurate approach for constructing of the dynamic modulus master curves for warm mix crumb rubber modified asphalt mixtures and assess the feasibility of predicting the phase angle master curves from the dynamic modulus ones. The SM (Sigmoidal model) and GSM (generalized sigmoidal model) were utilized to construct the dynamic modulus master curve, respectively. Subsequently, the master curve of phase angle could be predicted from the master curve of dynamic modulus in term of the K-K (Kramers–Kronig) relations. The results show that both SM and GSM can predict the dynamic modulus very well, except that the GSM shows a slightly higher correlation coefficient than SM. Therefore, it is recommended to construct the dynamic modulus master curve using GSM and obtain the corresponding phase angle master curve in term of the K-K relations. The Black space diagram and Wicket diagram were utilized to verify the predictions were consistent with the LVE (linear viscoelastic) theory. Then the master curve of storage modulus and loss modulus were also obtained. Finally, the creep compliance and relaxation modulus can be used to represent the creep and relaxation properties of warm-mix crumb rubber-modified asphalt mixtures.

## 1. Introduction

The mechanical behavior of viscoelastic materials is related to the frequency and loading history [1,2]. Asphalt material is a typical viscoelastic material, so viscoelastic behavior is essential for understanding road performance. Asphalt mixture will exhibit LVE (linear viscoelastic) properties in small strain levels or a limited number of cycles load [3,4,5,6]. The complex modulus test in oscillating load is the most conventional method for studying the LVE of asphalt mixtures. It can obtain the dynamic modulus and phase angle of the asphalt mixture in the LVE range. The dynamic modulus is easy to measure. However, due to the limitations of experimental equipment, the measurement of the phase angle will inevitably be scattered, and even in some cases, it is difficult to obtain by using the experimental method. Therefore, it is particularly important to establish the phase angle of the asphalt mixture derived from the results of dynamic modulus. Then it is vital to obtain the functional form of the dynamic modulus master curve. In the 1950s and 1960s, nomographs were widely used to represent the rheological properties of asphalt binder and mixtures [7]. Besides the standard logistic sigmoidal equation [8,9], the Weibull’s equations [10], and the generalized logistic sigmoid equation [11,12] were used to characterize the function form of dynamic modulus master curve for kinds of asphalt materials (binders or mixtures). Subsequently, Dickersen and Witt [13] proposed the relationship between phase angle, complex modulus, and frequency of asphalt binder. Christensen and Anderson [14] further simplified the above model and proposed the CA (Christensen–Anderson) model; Marasteanu and Anderson [15] proposed the CAM (Christensen–Anderson–Marasteanu) model in term of the CA model. Unfortunately, the model proposed by Dickersen and Witt only used independent parameters to establish the dynamic modulus master curves and phase angle master curve, respectively, which makes it difficult to satisfy the K-K (Kramers–Kronig) relations [16], and although other models ensure that the two kinds of master curves shared the same parameters, it is difficult to apply them for mixtures and binders. To ensure that the master curve meets the K-K relations, Booij and Thoone [17] first proposed that phase angles can be predicted by the slope of the complex modulus versus frequency based on the generalized K-K relations. This approach was also used by Christensen and Anderson [14] in obtaining phase angle of the Christensen Anderson model. Rowe [18] proposed a similar form of S-shaped equation for the analysis of asphalt mixture. Based on the K-K relations, Mensching [19] used the SM (sigmoidal model) and GSM (generalized sigmoidal model) to establish function form for the phase angle master curve and used the Black space parameters to evaluate the low-temperature performance of the asphalt mixture. Oshone [20] estimated phase angles based on dynamic modulus data for asphalt mixtures and evaluated the validity of the predictions by Black space diagram. Liu [21] also used the method to establish master curve model for the complex modulus of asphalt mixtures, which is highly accurate and consistent with the LVE theory. Nobakht [22] investigated the effects of aging on phase angle and dynamic modulus by performing complex modulus tests on asphalt mixtures aged at nine different levels in laboratory, the results showed that the phase angle prediction model based on the K-K relations could accurately predict the phase angle master curves after different aging. According to the principle of viscoelasticity, all of these research results show that the feasible to predict the phase angle master curve from the dynamic modulus. The purpose of this work is to evaluate the effectiveness of using different shifting techniques to establish the master curve model of dynamic modulus and phase angle for crumb rubber-modified asphalt mixtures by using the TTSP (time-temperature superposition principle) [23]. Besides, the functional form of the phase angle master curve can be predicted from the dynamic modulus master curve in term of the K-K relations [17,20]. And then, both the storage modulus master curve of and loss modulus master curve were obtained. Finally, the creep and relaxation properties of warm mix crumb rubber-modified asphalt mixtures (HMA) were investigated by the creep compliance and relaxation modulus, respectively.

## 2. Test Specimen Preparation and Testing

### 2.1. Materials and Specimen Fabricating Process

In this study, four kinds of laboratory-produced mixture were measured. Dense graded (AC-16) was utilized in the following text. Limestone for fine aggregates, and Basalt for coarse aggregates. Table 1 shows different aggregate stockpiles blended by the percentages, and Figure 1 shows the gradation of the aggregates. The performance grade of virgin asphalt is PG 64-22. The crumb rubber can be produced by mechanical grinding at ambient temperature. Besides, all of the crumb rubber prepared from the same source of waste bias tire. The crumb rubber modified asphalt binder was prepared by blending 20% of 60 (or compound) -mesh crumb rubber by the weight of virgin asphalt at 180 °C and 700 rpm/min for half an hour using a high-speed open blade mixer. Warm mix crumb rubber modified asphalt binder can be prepared by adding warm mix additive to the crumb rubber modified asphalt binder. The Surfactant (SDYK) was used as the warm mix additive, it was purchased from Wuxi Dowrid Chemical Technology Co (Wuxi, China). The content is 1% by the weight of asphalt binder (recommended by the manufacturer). It can be blended into the asphalt binder at 180 °C at 500 rpm for half an hour using a conventional mechanical mixer. Besides, the optimum asphalt content (determine by the Marshall method) is 5.4% and 5.6% by the weight of mixtures for hot mix 60-mesh crumb rubber-modified asphalt mixture (HMA-60) and hot mix compound-mesh crumb rubber-modified asphalt mixture (HMA-C), respectively. It can be learned from the Table 2 that the optimum asphalt content of HMA-C is greater than that of HMA-60. Because the former’s binder with higher viscous than the latter; thus, more binder is required to coat the aggregates. The viscosity test results of the crumb rubber modified bitumen binder can be obtained from Table A1 in the Appendix B. The optimum asphalt content of warm mix crumb rubber-modified asphalt mixture (WMA-60, WMA-C) is the same with the hot mix asphalt mixture. The compaction and mixing temperatures of HMA is 170 °C and 180 °C, respectively. While the compaction and mixing temperatures of WMA is at 152 °C and 162 °C, respectively. The target void content was 4% by the total volume. Table 2 shows the volumetric parameters of crumb rubber modified asphalt mixtures at optimum asphalt content. The virgin specimens (Ø150 mm × H178 mm) were prepared by Superpave gyratory compactor (IPC Global, Melbourne, Australia), then coring and sawing to produce the standard specimens (Ø100 mm × H150 mm).

### 2.2. Dynamic Modulus Test

The dynamic modulus test was conducted in a sinusoidal oscillating load, following in the AASHTO TP 79-15 [25]. The UTM-100 (closed-loop servo-hydraulic universal testing machine, IPC Global, Melbourne, Australia) was used to measure dynamic modulus and phase angle for all prepared specimens. The test was performed at seven frequencies (25 Hz, 20 Hz, 10 Hz, 5 Hz, 1 Hz, 0.5 Hz, 0.1 Hz) and four temperatures (5 °C, 20 °C, 35 °C, 50 °C). The axial strain was monitored by three LVDT (linear variable differential transducers) with a gauge length of 70 mm, which were mounted with 120 degree around the middle side of specimens. The average of three LVDT was retained as the axial strain. The load level was auto-adjusted to ensure that the maximum axial strain no more than 70 με, which was within the LVE domain [25]. Besides, the load was monitored by the load cell installed on the actuator. The raw data of dynamic modulus and phase angle for types of mixture is shown in Appendix A. Besides, the Correlation coefficient between the measurement and prediction of the dynamic modulus is shown in Appendix A.

## 3. Methodology

### 3.1. Shift Factors Calculated Methods

In the process of implementing the TTSP (time-temperature superposition principle), various shifting techniques were used to model the time-temperature superposition principle for asphaltic materials. In this study, three different types of shift factor function, including the Arrhenius equation [26], the WLF (Williams-Landel-Ferry) equation [27], and the second-order polynomial equation [28,29] were established for the SM and the GSM, respectively.

Arrhenius equation developed using the Arrhenius apparent activation energy [26]. The apparent activation energy describes the minimum energy required before any intermolecular motion can occur. In general, it was used to calculate the shift factor below the glass-transition temperature ($T_{g}$). It is also one of the oldest functions used to explain the relationship between viscoelastic properties and temperature. Equation (1) is the Arrhenius equation.
(1)$$\mathrm{lg}\alpha_{T}=\frac{\Delta E_{a}}{2.303R}\left(\frac{1}{T+273.15}-\frac{1}{T_{r}+273.15}\right)$$ where: $\alpha_{T}$ is the shift factor;  R is the mole gas constant, it is equal to 8.314 J·mol^−1^·K^−1^; ${\Delta E}_{a}$ is the apparent activation energy of the material, J/mol; T and $T_{r}$ are the experimental temperature and the reference temperature, respectively, °C.

Williams, Landel, and Ferry have constructed mathematical expressions between the shift factor and temperature based on free volume theory. That is the WLF (Williams-Landel-Ferry) equation [27]. It is noted that the WLF equation is used to characterize the relationship between the shift factor and temperature over a wide range ($T_{g}$~$T_{g}$ + 100), whether asphalt or mixtures. The function form was shown in Equation (2).
(2)$$\mathrm{lg}\alpha_{T}=\frac{-C_{1}\left(T-T_{r}\right)}{C_{2}+\left(T-T_{r}\right)}$$ where: $C_{1}$ and $C_{2}$ are constants determined by the thermodynamic properties of the material, which is used as fitting parameters in practical applications.

The second-order polynomial is also used to characterize function form between the shift factor and temperature. The functional form was shown in Equation (3).
(3)$$\mathrm{lg}\alpha_{T}=a{\left(T-T_{r}\right)}^{2}+b\left(T-T_{r}\right)$$ where: a, b are fitting constants which depend on the material properties and reference temperature.

### 3.2. Master Curve Model of the Dynamic Modulus

The magnitude of the dynamic modulus can be obtained from the complex modulus test. Complex modulus is the intrinsic material property of asphalt mixture. It is made up of the real part and the imaginary part. The real part represents elastic properties, called storage modulus, and the imaginary part represents viscous properties, called loss modulus [30]. The complex modulus can be estimated from the ratio of the stress input to the strain response, as shown in Equation (4). The dynamic modulus is the absolute value of the complex modulus. It can be calculated by the ratio of amplitude stress to amplitude strain, as shown in Equation (5). The dynamic modulus reflects the strength characteristics of the asphalt mixture. The phase angle reflects the time lagging between stress and strain. It can be calculated by the last five loading cycles. The function form can be obtained following the Equation (6) [31]. In general, the phase angle of elastic materials is equal to 0°. The phase angle of viscous materials is equal to 90°. the phase angle of viscoelastic materials varies from 0° to 90° [32,33].
(4)$$E^{*}=\frac{\sigma_{0}e^{i\left(wt+\varphi \right)}}{\varepsilon_{0}e^{iwt}}=E^{\prime }+iE^{\prime \prime }=\frac{\sigma_{0}}{\varepsilon_{0}}\cos\varphi +i\frac{\sigma_{0}}{\varepsilon_{0}}\sin\varphi$$
(5)$$\left|E^{*}\right|=\sqrt{{\left(E^{\prime }\right)}^{2}+{\left(E^{\prime \prime }\right)}^{2}}=\frac{\sigma_{0}}{\varepsilon_{0}}$$
(6)$$\varphi =\mathrm{actan}\frac{E^{\prime \prime }}{E^{\prime }}=\frac{t_{i}}{t_{p}}\times 360$$
where: $E^{*}$ is the complex modulus, MPa; $\sigma_{0}$ is the axial stress amplitude measured by load cell installed on the actuator, MPa; $\varepsilon_{0}$ is the axial strain amplitude measured by LVDT; i is the imaginary unit defined by $i=\sqrt{-1}$; $E^{\prime }$ is the storage modulus, MPa; $E^{\prime \prime }$ is the loss modulus, MPa; $t_{i}$ is the average time lag between the stress and strain for the last five loading cycles, s; $t_{p}$ is the average time of stress cycles for the last five loading cycles, s; φ is the phase angle, °.

Dynamic modulus data measured at multiple temperatures and frequencies were horizontal shifted to establish a single smooth and continuous master curve at arbitrary reference temperature to predict the viscoelastic behaviour of asphalt mixtures. The theoretical base for constructing the master curve of the dynamic modulus is TTSP. The master curve can be used to analyze the dynamic mechanical properties of the asphalt mixture at different loading frequencies and temperatures. In this study, 20 °C was used as the reference temperature.

Extensive research [34,35,36,37] shows that the dynamic modulus is close to maximum modulus value as the loading frequency increases to infinity and close to a limiting equilibrium value as the loading frequency approaches zero. Sigmoidal function and the generalized sigmoidal function can simulate the above characteristics of asphalt mixtures very well, called the SM and GSM, respectively. In this study, these two master curve models were used to fit the dynamic modulus obtained from different testing temperatures.

Equation (7) is the SM of the dynamic modulus master curve.
(7)$$\mathrm{lg}\left|E^{*}\right|=\delta +\frac{\alpha }{1+e^{\beta +\gamma \left(\log f_{r}\right)}}$$
where: $\left|E^{*}\right|$ is the predicted result of the dynamic modulus obtained from the SM, MPa; δ is the lower asymptote of the $\left|E^{*}\right|$ master curve in logarithmic coordinates; α is the vertical span between the lower and upper asymptotes of the $\left|E^{*}\right|$ master curve in logarithmic coordinates; β, γ is shape coefficients of the $\left|E^{*}\right|$ master curve, γ affects the rate of change between the lower asymptote and upper asymptotes, β affects the horizontal position of the turning point; $f_{r}$ is reduced frequency, Hz.

Compared with SM, the GSM added the parameter λ to characterize asymmetric characteristics [11,12]. It was presented as Equation (8).
(8)$$\mathrm{lg}\left|E^{*}\right|=\delta^{\prime }+\frac{\alpha^{\prime }}{{\left(1+\lambda e^{\beta^{\prime }+\gamma^{\prime }\left(\mathrm{lg}f_{r}\right)}\right)}^{\frac{1}{\lambda }}}$$
where: $\left|E^{*}\right|$ is the predicted result of the dynamic modulus obtained from the GSM, MPa; $\delta^{\prime }$ is the lower asymptote of the $\left|E^{*}\right|$ master curve in logarithmic coordinates; $\alpha^{\prime }$ is the vertical span between the upper and lower asymptotes of the $\left|E^{*}\right|$ master curve in logarithmic coordinates; $\beta^{\prime }$, $\gamma^{\prime }$ is shape parameters of the $\left|E^{*}\right|$ master curve obtained from the GSM.

The reduced frequency $f_{r}$ is the equivalent frequency of the experimental temperature with respect to the reference temperature. Moreover, the reduced frequency can be obtained by Equation (9) once the shift factor is obtained.
(9)$$\mathrm{lg}f_{r}=\mathrm{lg}f+\mathrm{lg}\alpha_{T}$$
where: lgf is the frequency in experiment temperature;$\;lgf_{r}$ is the reduced frequency in reference temperature.

### 3.3. Master Curve of the Phase Angle

Booij and Thoone [17] proposed that the function form of phase angle could be predicted by the relationship of complex modulus against frequency, as Equation (10). However, the Equation (10) has immense difficulties. For easy calculation, the equation was simplified appropriately, and the simplified equation was described as Equation (11), which is in term of the approximate K-K relations [38,39]. Finally, the function form of phase angle can be obtained by the SM and the GSM in term of the approximate K-K relations. The results were presented in Equations (12) and (13), respectively.
(10)$$\phi \left(f_{r}\right)=\frac{2f_{r}}{\pi }\int_{0}^{+\infty }\frac{\ln\left|E^{*}\left(u\right)\right|}{u^{2}-f_{r}{}^{2}}du$$
(11)$$\phi \left(f_{r}\right)=\frac{\pi }{2}{\left(\frac{d\ln\left|E^{*}\left(u\right)\right|}{d\ln u}\right)}_{u=f_{r}}=\frac{\pi }{2}{\left(\frac{d\mathrm{lg}\left|E^{*}\left(u\right)\right|}{d\mathrm{lg}u}\right)}_{u=f_{r}}$$
(12)$$\phi^{\prime }\left(f_{r}\right)=\frac{\pi }{2}\frac{d\left(\mathrm{lg}\left|E^{*}\right|\right)}{d\left(\mathrm{lg}f_{r}\right)}=-\frac{\pi }{2}\alpha \gamma \frac{e^{\beta +\gamma \mathrm{lg}f_{r}}}{{\left(1+e^{\beta +\gamma \mathrm{lg}f_{r}}\right)}^{2}}$$
(13)$$\phi^{\prime \prime }\left(f_{r}\right)=\frac{\pi }{2}\frac{d\left(\mathrm{lg}\left|E^{*}\right|\right)}{d\left(\mathrm{lg}f_{r}\right)}=-\frac{\pi }{2}\alpha^{\prime }\gamma^{\prime }\frac{e^{\beta^{\prime }+\gamma^{\prime }\mathrm{lg}f_{r}}}{{\left(1+\lambda e^{\beta^{\prime }+\gamma^{\prime }\mathrm{lg}f_{r}}\right)}^{1+\frac{1}{\lambda }}}$$
where: $\phi^{\prime }\left(f_{r}\right)$ is the phase angle predicted from the SM, °; $\phi^{\prime \prime }\left(f_{r}\right)$ is the phase angle predicted from the GSM, °; u is the integral variable; the other parameters are consistent with previously defined.

### 3.4. Determination of the Master Curve Model Parameters of Dynamic Modulus and Phase Angle

The master curves of dynamic modulus and phase angle for mixtures were constructed according to the SM and GSM. Three kinds of shift factor techniques were used to apply the TTSP to constructing the master curve model. There are five unknown model parameters ($\delta^{\prime }$, $\alpha^{\prime }$, $\beta^{\prime }$, $\gamma^{\prime }$, λ) for the generalized sigmoidal dynamic modulus master curve model and four unknown parameters (δ, α, β, γ) for the sigmoidal dynamic modulus master curve model. Besides, shift factor parameters were also calculated. The error function ef was applied to the results of dynamic modulus and phase angle to solve Equations (7) and (12) for the SM and Equations (8) and (13) for the GSM. The error function ef was demonstrated as Equation (14). All of the model parameters and the shift factors parameters were obtained by using the Microsoft solver function [40]. Finally, all parameter results (model fitting parameters and shift factors parameters) were showed in Table 3, Table 4, Table 5, Table 6, Table 7 and Table 8.
(14)$$ef=ef_{E^{*}}+ef_{\phi }=\frac{1}{N}\sqrt{\overset{N}{\underset{i=1}{\sum }}{\left(\frac{{\left|E^{*}\right|}_{m,i}-{\left|E^{*}\right|}_{p,i}}{{\left|E^{*}\right|}_{m,i}}\right)}^{2}}+\frac{1}{N}\sqrt{\overset{N}{\underset{i=1}{\sum }}{\left(\frac{{\left|\phi \right|}_{m,i}-{\left|\phi \right|}_{p,i}}{{\left|\phi \right|}_{m,i}}\right)}^{2}}$$
where: ef is the error function of dynamic modulus and phase angle; N is equal to 28, which is the number of measured samples (dynamic modulus or phase angle); ${\left|E^{*}\right|}_{m,i}$ is ith sample of dynamic modulus it can be obtained by measuring, MPa; ${\left|E^{*}\right|}_{p,i}$ is ith sample of dynamic modulus it can be predicted by dynamic modulus master curve, MPa; ${\left|\phi \right|}_{m,i}$ is ith sample of phase angle it can also be obtained by measuring, °; ${\left|\phi \right|}_{p,i}$ ith sample of phase angle it can be predicted by phase angle master curve, °.

From these tables (Table 3, Table 4, Table 5, Table 6, Table 7 and Table 8), it can be drawn that the model’s estimated value of dynamic modulus and phase angle are in good agreement with the experimental results. These results indicate the reliability of these model.

## 4. Results and Discussion

### 4.1. Comparison of the Shift Factors

Substitute the fitting parameters of Table 3, Table 4, Table 5, Table 6, Table 7 and Table 8 into the shift factor calculation equation to obtain the shift factor. For comparison purposes, all shift factor results of each mixture are shown in the same figure. Figure 2, Figure 3, Figure 4 and Figure 5 presents the shift factor results of four types of mixtures. These figures show no significant difference among the different shift factor techniques, especially for WLF equation and second-order polynomial equations. The specific manifestation is that the shift factor calculated by the Arrhenius equation is always smaller than the WLF equation and the second-order polynomial equation, and the higher temperature, the more significant. It is mainly due to the apparent activation energy being approximated as a temperature-independent constant in lower temperatures. However, in higher temperatures, it is the function of the temperature [41]. In other words, the Arrhenius equation is more suitable for use below the glass-transition temperature. Besides, the Arrhenius equation has only one freedom degree. However, the WLF equation and the second-order polynomial equation have multiple freedom degrees. The more freedom degrees, and better fitting.

### 4.2. Comparison of Dynamic Modulus Master Curve

#### 4.2.1. Sigmoidal Dynamic Modulus Master Curve

The sigmoidal dynamic modulus master curve model of HMA-60 and WMA-C was shown in Figure 6 and Figure 7. It was established using the WLF equation, the second-order polynomial equation, and the Arrhenius equation to calculate the shift factor, respectively. Compared with the experimental results of the laboratory, the master curves obtained by the first two methods showed a higher correlation. Although the sigmoidal dynamic modulus master curve model established by the Arrhenius shift factor calculation method has good agreement with the experimental results, the goodness of fitting is not as good as the first two methods, especially for HMA-60. It is because the shift factor calculated using the Arrhenius equation is always smaller than the WLF equation and the second-order polynomial equation. Compared with the dynamic modulus measured, the sinusoidal dynamic modulus master curve obtained by using the Arrhenius shift techniques is always slightly overestimated. Besides, the lower frequency, the more significant. Considering the paper’s page length limitations, only HMA-60 and WMA-C were selected for the illustration.

#### 4.2.2. Generalized Sigmoidal Dynamic Modulus Master Curve

Figure 8 and Figure 9 illustrated the result of the generalized sigmoidal dynamic modulus master curve model. It was also established by using the WLF equation, the second-order polynomial equation, and the Arrhenius equation to calculate the shift factor, respectively. The predicted value of each method shows a higher correlation with the experimental results of the laboratory, and there is almost no significant difference in the predicted results of the generalized sigmoidal master curve calculated by the first two shift factor methods. Although the generalized sigmoidal dynamic modulus master curve model established by the Arrhenius shift factor calculation method is also in good agreement with the experimental test results, the goodness of fitting statistic is not as good as that of the first two methods, especially for HMA-60. The primary reason for this is that the Arrhenius equation has only one freedom degree. However, the WLF equation and the second-order polynomial equation have multiple freedom degrees, the more freedom degrees, the better fitting. Considering the page length limitations of the paper, only two types of asphalt mixtures (HMA-60 and WMA-C) were selected for the illustration.

#### 4.2.3. Compared Sigmoidal Dynamic Modulus Master Curve and Generalized Sigmoidal Dynamic Modulus Master Curve

Figure 10 shows that there is no significant difference between the master curve model of sigmoidal and generalized sigmoidal constructed according to the Arrhenius equation. Besides, the predicted values of these two master curve models are significantly different from the experimental results of the laboratory in the low-frequency region. The figure also shows the predicted values of the sigmoidal dynamic modulus master curve and the generalized sigmoidal dynamic modulus master curve constructed by the WLF equation and the second-order polynomial equation. These results are consistent with the experimental results of the laboratory, whether low-frequency or high-frequency. It can also be learned from the figure that the master curve constructed in term of the Arrhenius equation cannot provide a good characterization in the low-frequency region. Therefore, it is not suitable for estimating the dynamic modulus in the low-frequency region. On the contrary, the dynamic modulus master curve model based on the WLF equation and the second-order polynomial equation can accurately predict the dynamic modulus at all frequencies. Compared with the experimental results of the laboratory, the generalized sigmoidal dynamic modulus master curve has better goodness of fitting than the sigmoidal dynamic modulus master curve. This is because it can characterize the asymmetry of the dynamic modulus master curve. The result is consistent with Yusoff’s work [42]. Due to page’s length limitations, only HMA-60 is selected for presenting.

### 4.3. Comparison of Phase Angle Master Curve

According to Equations (12) and (13), the slope method can be used to predict the function form of phase angle from the dynamic modulus on the basis of K-K relations. In this paper, the functional form between the dynamic modulus and frequency has been characterized previously using the sigmoidal master curve model and the generalized sigmoidal master curve model, respectively. Finally, the predicted results of the phase angle can be obtained by substituting the parameter results of Table 3, Table 4, Table 5, Table 6, Table 7 and Table 8 in Equations (12) and (13), as detailed in Figure 11, Figure 12, Figure 13 and Figure 14.

#### 4.3.1. Prediction the Phase Angle Master Curve from Sigmoidal Dynamic Modulus Master Curve

Figure 11 gives the predictions of the phase angle master curve by using SM for the HMA-60. Considering that three kinds of shifting technique have been used to construct the sigmoidal dynamic modulus master curves, namely the WLF equation, the second-order polynomial equation, and the Arrhenius equation, then the phase angle master curve constructed based on the sigmoidal dynamic modulus master curve also has three predictions. As shown in Figure 11, the phase angle master curve predicted by the sigmoidal dynamic modulus master curve is consistent with the experimental results of laboratory. The phase angle master curve predicted by the sigmoidal dynamic modulus master curve constructed based on the Arrhenius shift technique is closer to the left than the other two methods. Besides, the phase angle master curve obtained based on the first two shift factor calculation methods have a better goodness of fitting than the Arrhenius equation. A similar observation is made for the WMA-C. The results are presented in Figure 12. Considering the paper’s page length limitations, only two types of asphalt mixtures, that is HMA-60 and WMA-C, were selected for the illustration.

#### 4.3.2. Prediction the Phase Angle Master Curve from Generalized Sigmoidal Dynamic Modulus Master Curve

Figure 13 gives the predicted results of the phase angle master curve for HMA-60. It can be obtained from the GSM based on the K-K relations. The predicted phase angle master curves have a good agreement with the experimental results of the laboratory. Similar to previous, the phase angle master curve predicted by the generalized sigmoidal dynamic modulus master curve constructed based on the Arrhenius shift technique is closer to the left than the other two methods. The phase angle master curves obtained from the WLF equation and the second-order polynomial equation have a better goodness of fitting than the Arrhenius equation. A similar observation is made for WMA-C. The results are shown in Figure 14. Considering the paper’s page length limitations, only two types of asphalt mixtures, that is HMA-60 and WMA-C, were selected for the illustration.

#### 4.3.3. Compared Phase Angle Master Curve Obtained by the Sigmoidal Dynamic Modulus Master Curve and Generalized Sigmoidal Dynamic Modulus Master Curve

Figure 15 shows the predicted results of the phase angle master curve for WMA-C by using the SM and GSM over a wide range of reduced frequency. As mentioned before, both the SM and GSM were constructed by using the Arrhenius equation, the WLF equation, and the second-order polynomial equation to calculate the shift factor. Compared with the laboratory’s experimental results, these phase angle predictions obtained based on SM and GSM show a high correlation, except that the latter has a slightly better prediction accuracy than the former. It was also proved that this method has a better application prospect, especially for the condition of lacking phase angle information [43] (for example, the long-term pavement performance). The phase angle master curve obtained by the WLF equation and the second-order polynomial equation has better goodness of fitting than the Arrhenius equation. The reason is that the Arrhenius equation has only one freedom degree. However, the WLF equation and the second-order polynomial equation have multiple freedom degrees. The more freedom degrees, the better fitting. Considering the page length limitations of the paper, only WMA-C, was selected for the illustration.

### 4.4. Verification of Compliance with LVE Theory between Dynamic Modulus and Phase Angle Master Curves

Plotting the dynamic modulus in logarithmic coordinates and the phase angle in arithmetic coordinates, then the Black space diagram [44] was obtained. Here, it was also used to assess the predicted results of the dynamic modulus and phase angle as well as monitor the results’ quality. Figure 16 presents the Black space diagrams of WMA-C. In the Black space diagram, the measured value and the predicted value from model curves were shown and compared. The predicted value of dynamic modulus obtained from the GSM and shift factors obtained from the WLF equation. The estimated value of the phase angle obtained from the GSM base on the K-K relations. The results show that all the test data lie on or stay close to a unique smooth curve, which verified that the constructed dynamic modulus and phase angle master curves were consistent with the LVE theory. As Figure 17, the results have also been proven on the Wicket diagram (storage modulus and loss factor were plotted against each other in a semi-log graph) [21]. Considering the limitation of the document, other types of asphalt mixtures are not demonstrated here.

Finally, the master curves of dynamic modulus and phase angle for the four kinds of crumb rubber-modified asphalt mixtures are plotted together in Figure 18, where the GSM is used to construct the dynamic modulus master curve, then the phase angle master curve could be obtained in term of the K-K relations. Besides, the shift factors could be obtained by using the WLF equation.

It can be observed in Figure 18 that the master curve of the dynamic modulus exhibits the S-shaped. It means that the crumb rubber-modified asphalt mixtures mainly characterize elastic in higher frequency (or low temperature) but are viscous in lower frequency (or high temperature). The master curve of the phase angle exhibits bell-shaped. It increased with the increase of frequency to the maximum phase angle and then decrease with the further increase of frequency. The phase angles are close to zero when the frequency close to zero or infinity, indicating that the crumb rubber-modified asphalt mixtures exhibit elastic characteristics at extremely low or high frequencies. This is because in high frequency (or low temperature), the asphalt mixture behaves as elastic in nature and is mainly subjected to asphalt binder, thus shows a higher dynamic modulus and smaller phase angle. When decreasing frequency (or increases temperature), the asphalt binder becomes soft and approach viscous, leading to the decreased dynamic modulus and increased phase angle. In lower frequency (or higher temperature), asphalt mixture mainly behaves as viscous, and the influence of the asphalt binder on the mixture becomes weak while the interlocking force between aggregates on mixture becomes distinct, then leading to the decrease dynamic modulus and phase angle. With the further decrease frequency (or increase temperature), the phenomenon where the aggregates skeleton mainly bears the loading stress becomes more obvious, then leading to a lower dynamic modulus and smaller phase angle [45]. However, the frequency dependence of phase angle for the crumb rubber-modified asphalt binder was not consistent with that of the corresponding mixture (see Figure A1 in Appendix B). It can be learned that the phase angle of the binder decreases monotonically with the frequency increases. When the frequency approach infinity, the phase angle close to a limiting equilibrium value; when the frequency approach 0, the phase angle close to 90°. It can be attributed to the significant influence of the aggregate at low frequencies.

Compared to the dynamic modulus master curves of the four types of mixtures, it learns that the dynamic modulus of HMA-60 always has smaller than that of HMA-C. However, once the warm mix additive was added, the dynamic modulus of the mixture in high-frequency becomes smaller and larger in low-frequency. The dynamic modulus of WMA-60 is always smaller than WMA-C, which shows that the strength of HMA-C is better than HMA-60. Once the warm mix additive was added, the strength of the mixture in low-frequency (high temperature) increased and decreased in high-frequency (low temperature), and the strength of WMA-C is better than that of WMA-60. Compared to the phase angles master curves of four types of mixtures, it is found that the phase angle of HMA-60 is always greater than that of HMA-C. Once the warm mix additive was added, the phase angle of the mixture at high-frequency becomes larger and smaller in low-frequency. From the above, it is clear that once the warm mix additive was added, the viscous flow of the asphalt mixture decreases in the low-frequency region (high temperature) and increases in the high-frequency region (low temperature). It is because the viscosity of the 60-mesh crumb rubber-modified asphalt binder is less than the compound-mesh crumb rubber-modified asphalt binder. Then the adhesion of the mixture produced from the former is less than the latter, which results in the 60-mesh crumb rubber-modified asphalt mixture with lower dynamic modulus and higher phase angle. Once warm mix additive was added, the mixing temperature of the mixture will be reduced. This process also reduces the aging of the asphalt binder. The fluidity of the asphalt binder was also improved, the aggregate can absorb more asphalt, and the content of the structural asphalt will increase, so the dynamic modulus of warm mix crumb rubber-modified asphalt mixture is greater than hot mix crumb rubber-modified asphalt mixture in the lower frequency range (high temperature).

### 4.5. Comparing Other LVE Response Function of the Asphalt Mixture

#### 4.5.1. Comparing the Storage Modulus and the Loss Modulus

The storage modulus and loss modulus can be calculated according to Equations (15) and (16), respectively, once knowing the dynamic modulus and phase angle at different frequencies and temperatures. The storage modulus master curve and loss modulus master curve are also established. The results are shown in Figure 19.
(15)$$E^{\prime }=\left|E^{*}\right|\cos\left(\phi \right)$$
(16)$$E^{\prime \prime }=\left|E^{*}\right|\sin\left(\phi \right)$$
where: $E^{\prime }$ is the Storage modulus, MPa; and $E^{\prime \prime }$ is the Loss modulus, MPa.

Figure 19 is the Storage modulus master curve and Loss modulus master curve for four kinds of asphalt mixture. It is obvious that the storage modulus master curve is similar to the dynamic modulus master curve, especially at the high frequencies range. As the frequency increases, the storage modulus increases gradually, and there is the minimum value in the low-frequency region and the maximum value in the high-frequency region. The master curve is a typical S-shape. This is because the phase angle of the mixture is very low, especially at high frequencies, where it is approach to a very small magnitude. The storage modulus of HMA-C is always greater than HMA-60. The mixture’s storage modulus in the low-frequency range increased and decreased in the high-frequency range after warm mix additive was added. The storage modulus of WMA-C is always greater than WMA-60. Obviously, the elastic deformation resistance of HMA-C is better than HMA-60. Once warm mix additive was added, the elastic deformation resistance could be improved in the low-frequency (high temperature) range. Moreover, the elastic deformation resistance of WMA-C is better than WMA-60.

Figure 19 also illustrates that the master curve of the loss modulus. It can be learned that the loss modulus first increases and then slightly decreases with the frequency increases. The loss modulus of HMA-C is always greater than HMA-60. Once warm mix additive was added, the loss modulus of HMA-60 the mixture in the low-frequency range becomes larger and in the high-frequency range becomes smaller, but the Loss modulus of HMA-C in the high-frequency range becomes larger and smaller in the low-frequency range. The loss modulus of WMA-C is larger than that of WMA-60 in the middle and low-frequency range. It showed that the viscous deformation resistance of HMA-C is better than HMA-60. Once warm mix additive was added, the viscous deformation resistance could be increase in the low-frequency range and decrease in the high-frequency range. Moreover, the viscous deformation resistance of WMA-C is better than WMA-60.

In summary, it can be learned that both elastic deformation resistance and viscous deformation resistance of HMA-C is better than HMA-60. Once the warm mix additive was added, the elastic deformation resistance and viscous deformation resistance of the mixture could be improved in high temperature; WMA-C shows a better deformation resistance than WMA-60. It is because the viscosity of the compound-mesh crumb rubber-modified asphalt binder is greater than 60-mesh crumb rubber-modified asphalt binder. The adhesion of the mixture produced from the former is greater than the latter, which results in the compound-mesh crumb rubber-modified asphalt mixture with higher strength and better resistance to deformation. Once warm mix additive was added, the mixing temperature of the mixture will be reduced. This process also reduces the aging of the asphalt binder. The fluidity of the asphalt binder was also improved, the aggregate can absorb more asphalt, and the content of the structural asphalt will increase, so the deformation resistance of warm mix crumb rubber-modified asphalt mixture is better than hot mix crumb rubber-modified asphalt mixture in high temperature. From the molecular point, the primary reason for the above results can be attributed to the decrease in molecular weight and the concentration of polar functional groups in the asphalt binders after the warm mix additive was added [46].

#### 4.5.2. Comparing to the Relaxation Modulus

Viscoelastic properties are inherent properties of asphalt materials. In the effect of temperature and vehicle load, the pavement structure’s internal stress will gradually dissipate with the increase of time, which is stress relaxation [47]. The stress accumulated in the pavement structure will quickly dissipate due to the strong relaxation ability in the higher temperature; However, when in the lower temperature, the relaxation ability of the asphalt pavement is weak, the relaxation speed is slow, and there is excessive stress accumulation in the pavement structure. Once the stress accumulation exceeds the strength of the mixture, the asphalt pavement will crack, so it is essential to study the asphalt mixture’s relaxation modulus.

According to the relation between complex material functions and operational functions [48], we can obtain the storage modulus and loss modulus expressed in the Prony series, and the expression is as Equations (17) and (18).
(17)$$E^{\prime }\left(w\right)=E_{e}+\overset{m}{\underset{i=1}{\sum }}\frac{w^{2}\rho_{i}^{2}E_{i}^{2}}{w^{2}\rho_{i}^{2}+1}$$
(18)$$E^{\prime \prime }\left(w\right)=\overset{m}{\underset{i=1}{\sum }}\frac{w\rho_{i}E_{i}}{w^{2}\rho_{i}^{2}+1}$$
where: $E_{e}$ is the long-term equilibrium modulus. It is equal to the dynamic modulus as the reduced frequency approach to zero, MPa; $E_{i}$ is the relaxation strengths of ith Maxwell component, MPa; $\rho_{i}$ is the relaxation time of ith Maxwell component, s; w is the reduced angular frequency and is equal to $2\pi f_{r}$, rad/s.

To obtain equilibrium modulus and relaxation strength of all Maxwell component. The error function $ef_{1}$ was applied to the test data of storage modulus and loss modulus to solve Equations (17) and (18) simultaneously. It was demonstrated as Equation (19), relaxation time taken with equidistant intervals on the logarithmic t axis [41] as Table 9. The detailed calculation process can be found in the results of Zhang [24]. Finally, the results for both the long-term equilibrium modulus and the relaxation strength are also presented in Table 9.
(19)$$ef_{1}=ef_{E^{\prime }}+ef_{E^{\prime \prime }}=\frac{1}{N}\sqrt{\overset{N}{\underset{i=1}{\sum }}{\left(\frac{{\left|E^{\prime }\right|}_{m,i}-{\left|E^{\prime }\right|}_{p,i}}{{\left|E^{\prime }\right|}_{m,i}}\right)}^{2}}+\frac{1}{N}\sqrt{\overset{N}{\underset{i=1}{\sum }}{\left(\frac{{\left|E^{\prime \prime }\right|}_{m,i}-{\left|E^{\prime \prime }\right|}_{p,i}}{{\left|E^{\prime \prime }\right|}_{m,i}}\right)}^{2}}$$
where: $ef_{1}$ is the error function of storing modulus and loss modulus; N is equal to 28, it is the number of measured samples; ${\left|E^{\prime }\right|}_{m,i}$ is ith data point of the storage modulus, it can be obtained by measurement, MPa; ${\left|E^{\prime }\right|}_{p,i}$ is ith data point of storage modulus, it can be predicted by Prony series, MPa; ${\left|E^{\prime \prime }\right|}_{m,i}$ is ith data point of the loss modulus, it can be obtained by measurement, MPa; ${\left|E^{\prime \prime }\right|}_{p,i}$ is ith data point of loss modulus, it can be predicted by Prony series, MPa.

Once the Equation (19) can be calculated, Equation (20) can demonstrate relaxation modulus. The function of relaxation modulus verse time was revealed in Figure 20.
(20)$$E\left(t\right)=E_{e}+\overset{m}{\underset{i=1}{\sum }}E_{i}e^{-\frac{t}{\rho_{i}}}$$
where: $E\left(t\right)$ is relaxation modulus, MPa.

As illustrated in Figure 20, The relaxation modulus of HMA-C greater than that of HMA-60. Once the warm mix additive was added, the relaxation modulus will decrease in the shorter time range and increase in the longer time range; the relaxation modulus of WMA-C is always greater than WMA-60. It showed that the relaxation ability of the HMA-60 is always better than HMA-C. Once the warm mix additive was added, it can improve the low-temperature relaxation ability, and the relaxation ability of WMA-60 always better than that of WMA-C. This is also because the viscosity of the 60-mesh crumb rubber-modified bitumen binder is less than that of the compound-mesh crumb rubber-modified bitumen binder. Then the mixture produces by using the latter binder with more pronounced elastic properties, which resulting in the compound-crumb rubber-modified asphalt mixture with poor relaxation characteristics. Once the warm mix additive was added, the mix temperature of the mixture will be reduced. This process also reduced the aging of the asphalt mixture. In addition, it also improves the fluidity of the mixtures, so the warm-mix crumb rubber-modified asphalt mixture with better relaxation characteristics than the hot-mix crumb rubber-modified asphalt mixture in lower time range (low temperature).

#### 4.5.3. Comparing to the Creep Compliance

Similarly, according to the relation between complex material functions and operational functions. The storage compliance $D^{\prime }\left(w\right)$ and loss compliance $D^{\prime \prime }\left(w\right)$ can be determined by the interconversion between the complex modulus and complex compliance as Equation (21). Then storage compliance and loss compliance can be rewrite in the Prony series as Equations (22) and (23).
(21)$$D^{*}=\frac{1}{E^{*}}=D^{\prime }-iD^{\prime \prime }=\frac{E^{\prime }}{E^{\prime 2}+E^{\prime \prime 2}}-i\frac{E^{\prime \prime }}{E^{\prime 2}+E^{\prime \prime 2}}$$
(22)$$D^{\prime }\left(w\right)=D_{g}+\sum_{j=1}^{n}\frac{D_{j}}{w^{2}\tau_{j}^{2}+1}$$
(23)$$D^{\prime \prime }\left(w\right)=\frac{1}{\eta_{0}w}+\sum_{j=1}^{n}\frac{w\tau_{j}D_{j}}{w^{2}\tau_{j}^{2}+1}$$
where: $D_{g}$ is the long-term equilibrium compliance, It is equal to the reciprocal of the dynamic modulus as the reduced frequency approach to zero, MPa^−1^; $\eta_{0}$ is the zero shear or long time viscosity (for asphalt mixture $\eta_{0}=\infty$), cP; $D_{j}$ is the retardation strengths of ith Kelvin component, MPa^−1^; $\tau_{j}$ is the retardation times of ith Kelvin component, s; w is the reduced angular frequency and it is equal to $2\pi f_{r}$, rad/s.

To obtain equilibrium compliance and retardation strength of all Kelvin component. The error function $ef_{2}$ was applied to the test data of storage compliance and loss compliance to solve Equations (22) and (23) simultaneously. It was demonstrated as Equation (24), the retardation time can be obtained from Park and Schapery method [48], and the results were shown in Table 10. Similarly, the detailed calculation process can be found in the results of Zhang [24]. Finally, the results for both the long-term equilibrium compliance and the retardation strengths are also presented in Table 10.
(24)$$ef_{2}=ef_{D^{\prime }}+ef_{D^{\prime \prime }}=\frac{1}{N}\sqrt{\overset{N}{\underset{i=1}{\sum }}{\left(\frac{{\left|D^{\prime }\right|}_{m,i}-{\left|D^{\prime }\right|}_{p,i}}{{\left|D^{\prime }\right|}_{m,i}}\right)}^{2}}+\frac{1}{N}\sqrt{\overset{N}{\underset{i=1}{\sum }}{\left(\frac{{\left|D^{\prime \prime }\right|}_{m,i}-{\left|D^{\prime \prime }\right|}_{p,i}}{{\left|D^{\prime \prime }\right|}_{m,i}}\right)}^{2}}$$
where: $ef_{2}$ is the error function of storing compliance and loss compliance; N is equal to 28, which is the number of measured samples; ${\left|D^{\prime }\right|}_{m,i}$ is *i*th data point of the storage compliance, it can be obtained by measurement, MPa^−1^; ${\left|D^{\prime }\right|}_{p,i}$ is ith data point of storage compliance, it can be predicted by Prony series, MPa^−1^; ${\left|D^{\prime \prime }\right|}_{m,i}$ is ith data point of the loss compliance, it can be obtained by measurement, MPa^−1^; ${\left|D^{\prime \prime }\right|}_{p,i}$ is ith data point of the loss compliance, it can be predicted by Prony series, MPa^−1^.

It is clear that, once Equation (24) can be calculated, then Equation (25) can be used for demonstrating creep compliance $D\left(t\right)$. The function of creep compliance against time was also shown in Figure 20.
(25)$$D\left(t\right)=D_{g}+\frac{t}{\eta_{0}}+\overset{n}{\underset{j=1}{\sum }}D_{j}\left(1-e^{-\frac{t}{\tau_{j}}}\right)$$

Figure 20 also shows that the creep compliance of HMA-C is always less than HMA-60. Once the warm mix additive was added, the mixture’s creep compliance will increase in the shorter time range and decrease in the longer time range. The creep compliance of WMA-C is always less than WMA-60. It showed that the deformation resistance of the HMA-C is always better than HMA-60. Once the warm mix additive was added, it can improve the deformation resistance ability at high temperatures, and the deformation resistance of the WMA-C is always better than WMA-60.

## 5. Conclusions

This study involves constructing SM and GSM for dynamic modulus of crumb rubber-modified asphalt mixtures using different shifting techniques. The phase angle can be obtained from the dynamic modulus master curve by K-K relations. The Black space diagram and Wicket diagram were used to evaluate the predicted results of the dynamic modulus and phase angle. Finally, the relaxation modulus and creep compliance can be used to characterize the relaxation and creep properties of warm mix crumb rubber-modified asphalt mixture. Based on the results of this study, the following conclusions were drawn:
(1)The shift factor calculated by the Arrhenius equation is always smaller than the WLF equation and the second-order polynomial equation, and the higher temperature, the more significant.(2)Both SM and GSM can be used as the master curve models of dynamic modulus, except that GSM presents slightly excellent fitting than SM.(3)Compared with the laboratory results, the prediction of phase angles constructed based on the K-K relations shows a higher correlation coefficient. Moreover, the accuracy of the predicted phase angle depends on the accuracy of the dynamic modulus master curve.(4)The Black space diagram and the Wicket diagram demonstrate that the master curve of dynamic modulus and phase angle is constructed by the slope method compliance LVE theory.(5)According to the viscoelastic theory, the storage modulus master curve and the loss modulus master curve can be obtained from the complex modulus test. Furthermore, the storage compliance master curve and the loss compliance master curve can also be obtained. Finally, the master curve of the relaxation modulus and creep compliance can be obtained in the region.(6)From the results of dynamic modulus and phase angle, we can obtain that the deformation resistance of HMA-60 is not as good as HMA-C. Once the warm mix Additive was added, the mixture’s deformation resistance in the low-frequency region (high temperature) will be improved, the viscous flow in the high-frequency region (low temperature) will also be enhanced. The WMA-C presents a better deformation resistance at high temperature, while WMA-60 presents better crack resistance at low temperature.(7)From the results of relaxation modulus and creep compliance, it can be learned that the HMA-60 exhibits better low-temperature deformation but less high-temperature deformation resistance than the HMA-C. In addition, the WMA exhibits better low-temperature deformation and high-temperature deformation resistance than the corresponding HMA.


## Figures and Tables

**Figure 1 materials-13-05051-f001:**
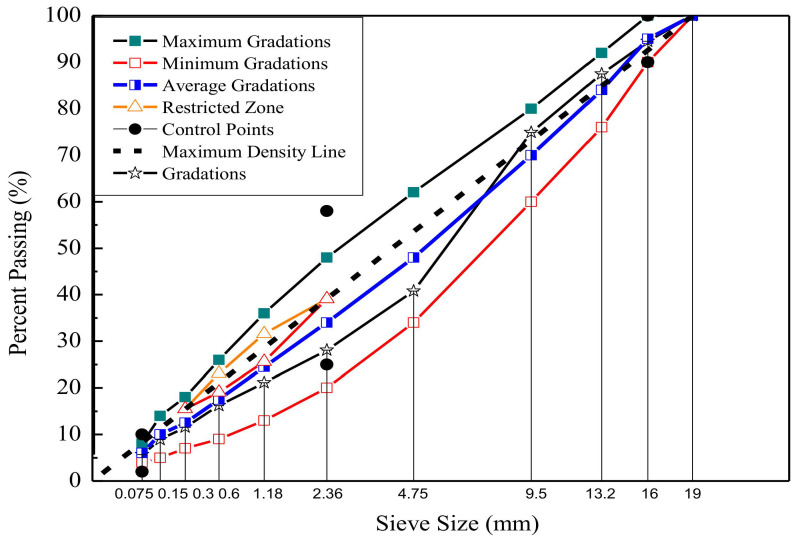
The gradation of crumb rubber-modified asphalt mixtures [24].

**Figure 2 materials-13-05051-f002:**
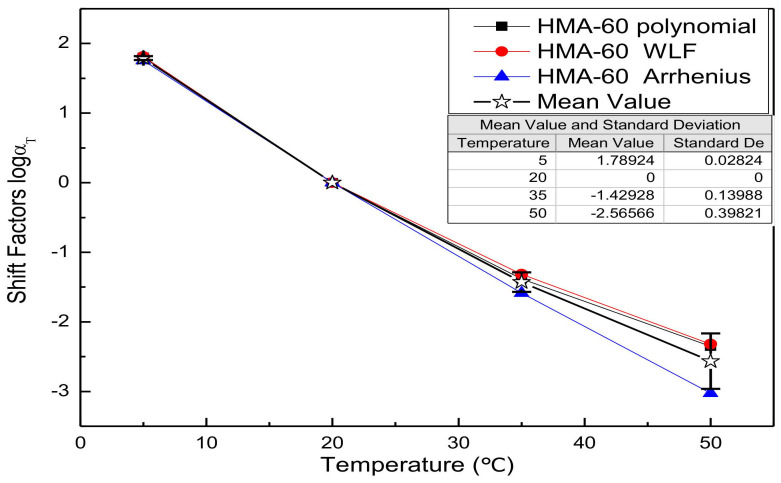
Shift factors of HMA-60.

**Figure 3 materials-13-05051-f003:**
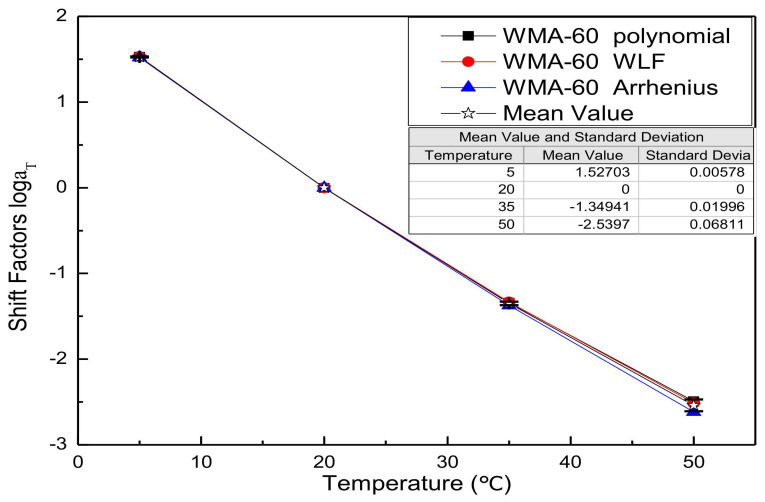
Shift factors of WMA-60.

**Figure 4 materials-13-05051-f004:**
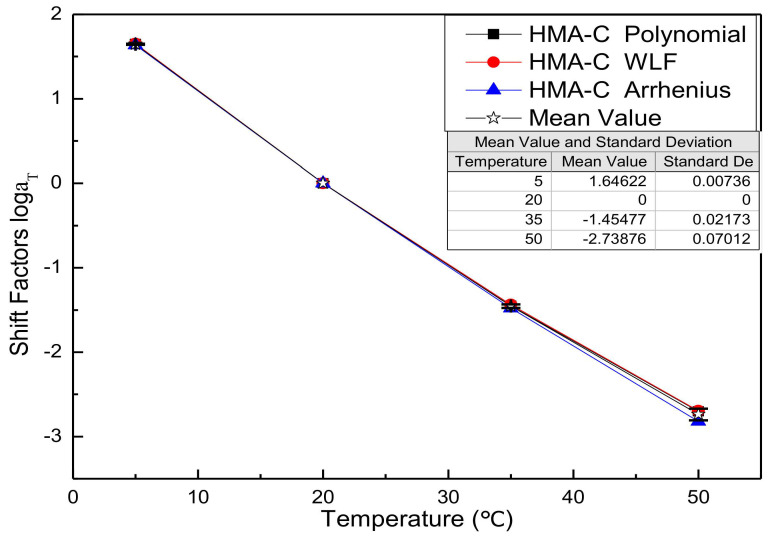
Shift factors of HMA-C.

**Figure 5 materials-13-05051-f005:**
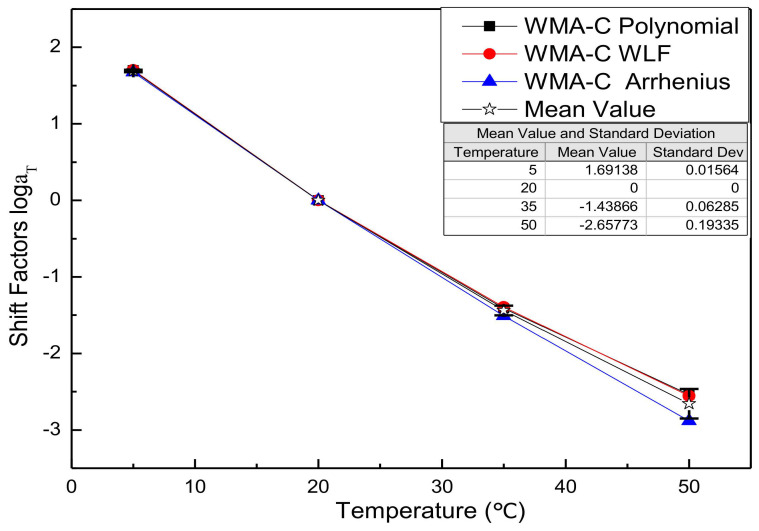
Shift factors of WMA-C.

**Figure 6 materials-13-05051-f006:**
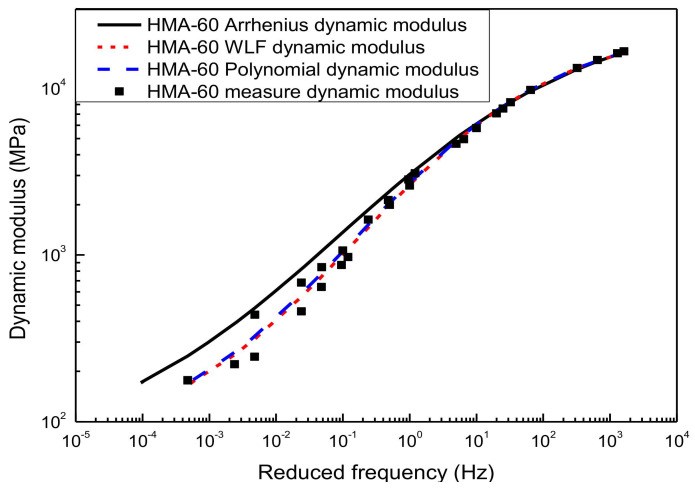
Sigmoidal dynamic modulus master curve of HMA-60.

**Figure 7 materials-13-05051-f007:**
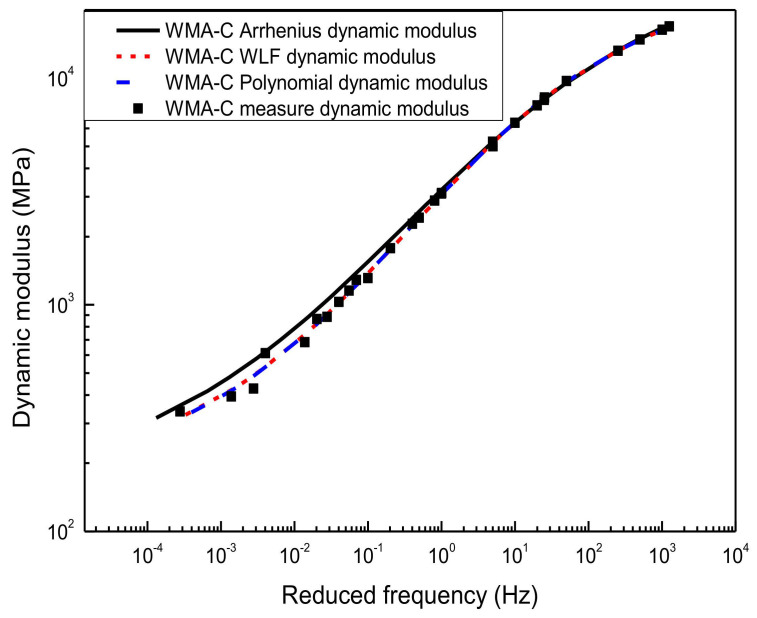
Sigmoidal dynamic modulus master curve of WMA-C.

**Figure 8 materials-13-05051-f008:**
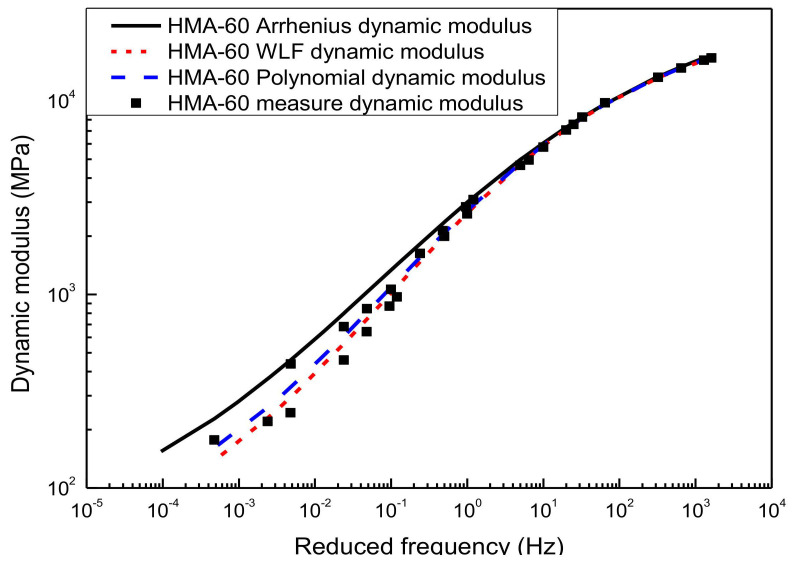
Generalized sigmoidal dynamic modulus master curve of HMA-60.

**Figure 9 materials-13-05051-f009:**
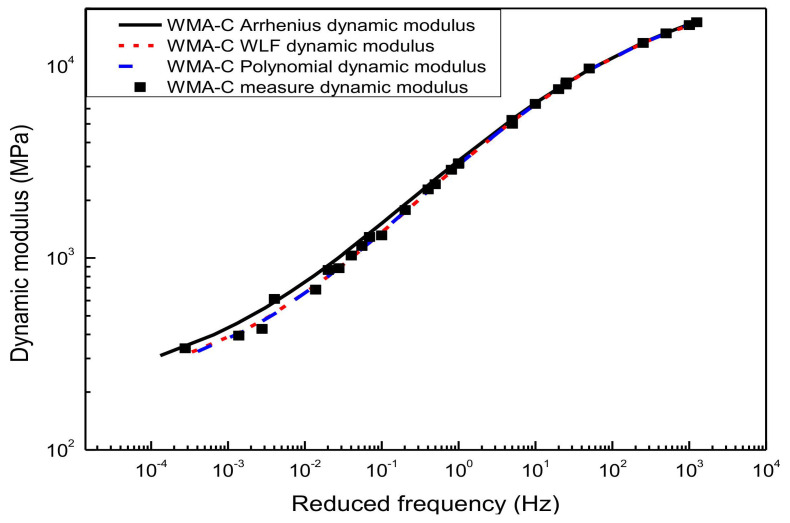
Generalized sigmoidal dynamic modulus master curve of WMA-C.

**Figure 10 materials-13-05051-f010:**
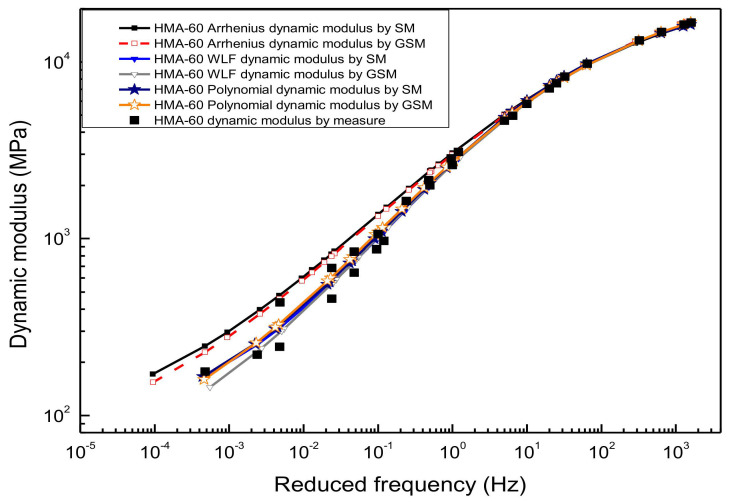
Compared the dynamic modulus master curve obtained by different methods.

**Figure 11 materials-13-05051-f011:**
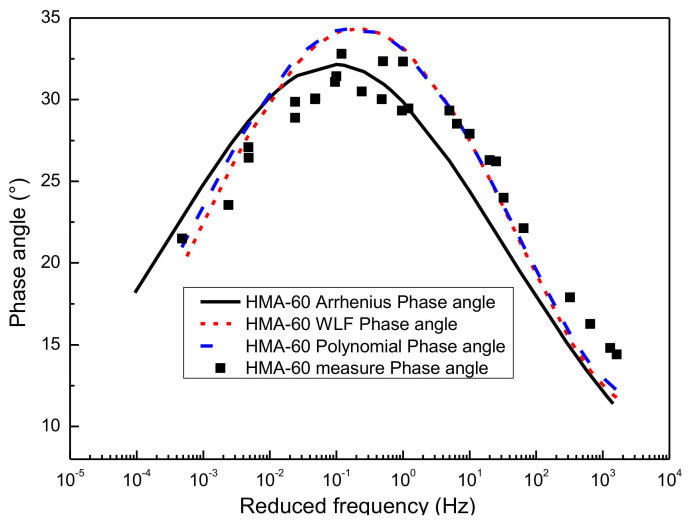
Predict phase angle master curve of HMA-60 from SM.

**Figure 12 materials-13-05051-f012:**
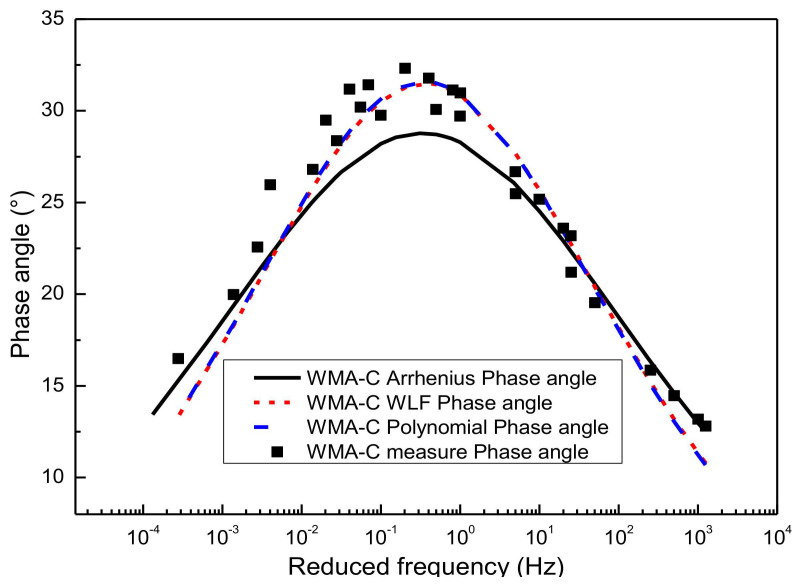
Predict phase angle master curve of WMA-C from SM.

**Figure 13 materials-13-05051-f013:**
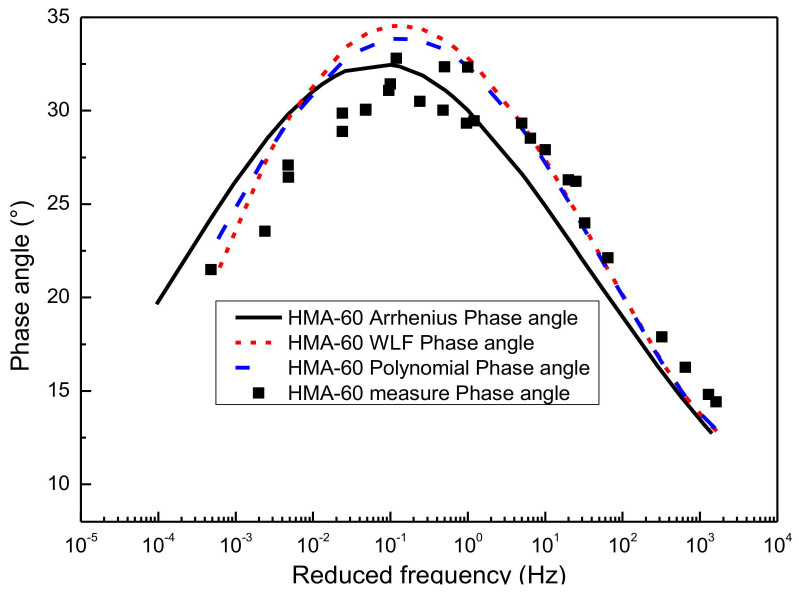
Predict phase angle master curve of HMA-60 from generalized sigmoidal.

**Figure 14 materials-13-05051-f014:**
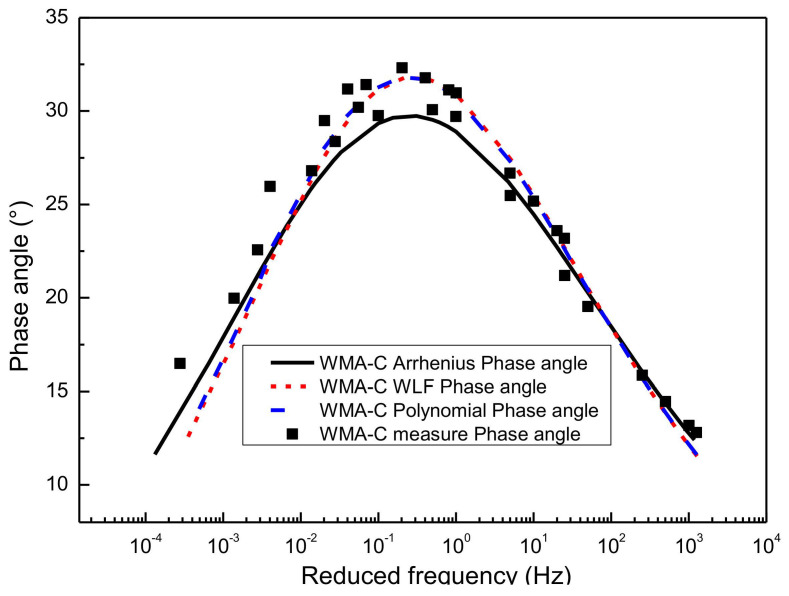
Predict phase angle master curve of WMA-C from generalized sigmoidal.

**Figure 15 materials-13-05051-f015:**
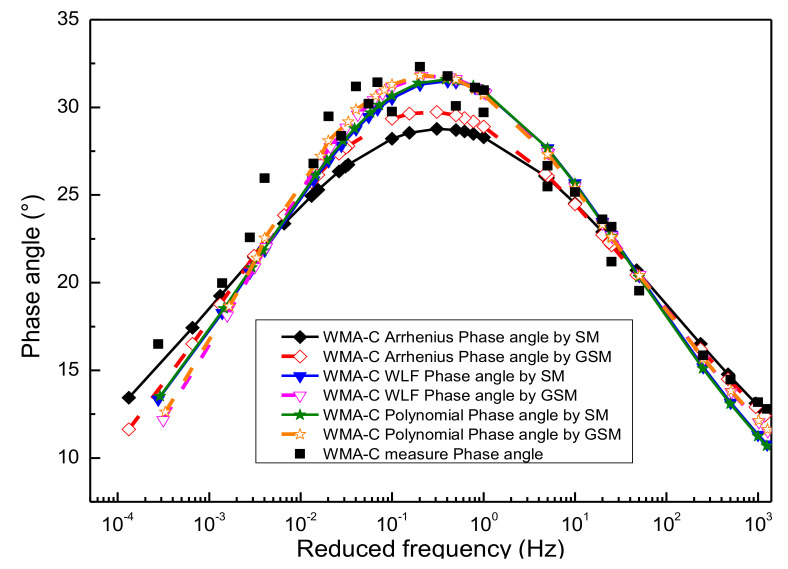
Compared phase angle master curve obtained by different methods.

**Figure 16 materials-13-05051-f016:**
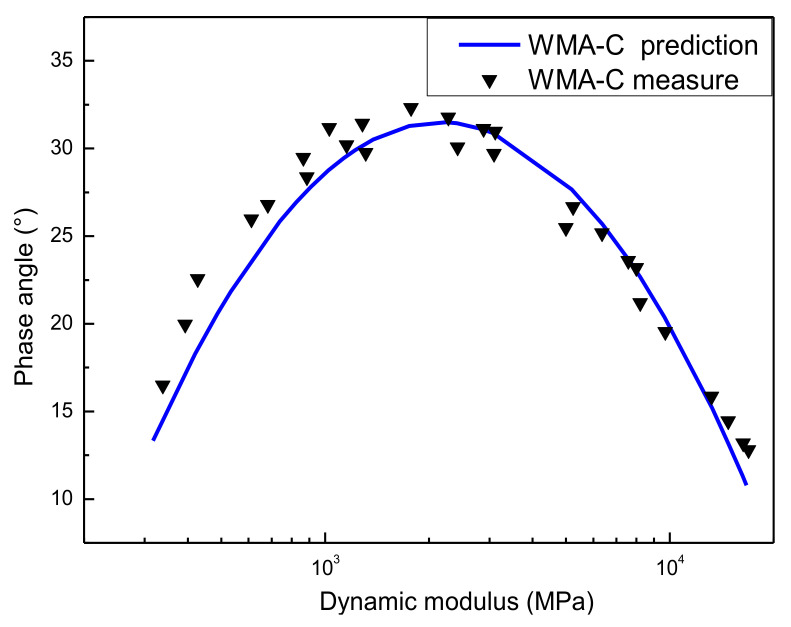
The Black space diagram of the asphalt mixture.

**Figure 17 materials-13-05051-f017:**
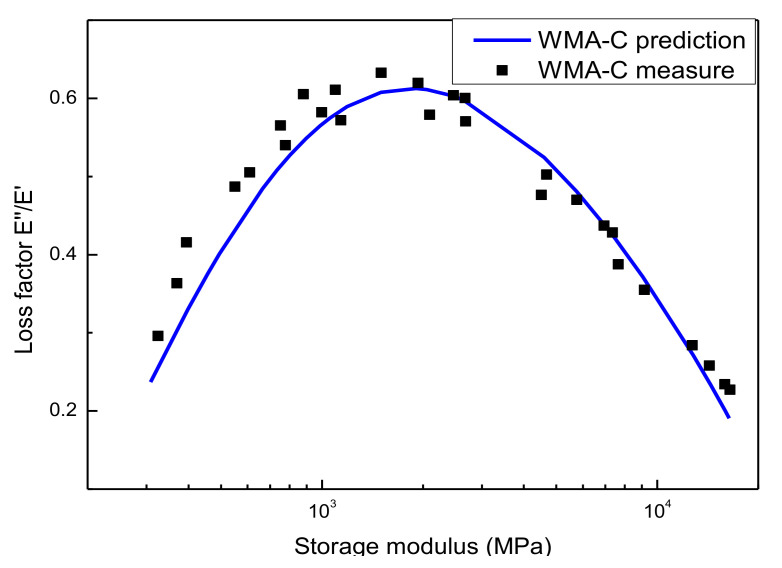
The Wicket diagram of the asphalt mixture.

**Figure 18 materials-13-05051-f018:**
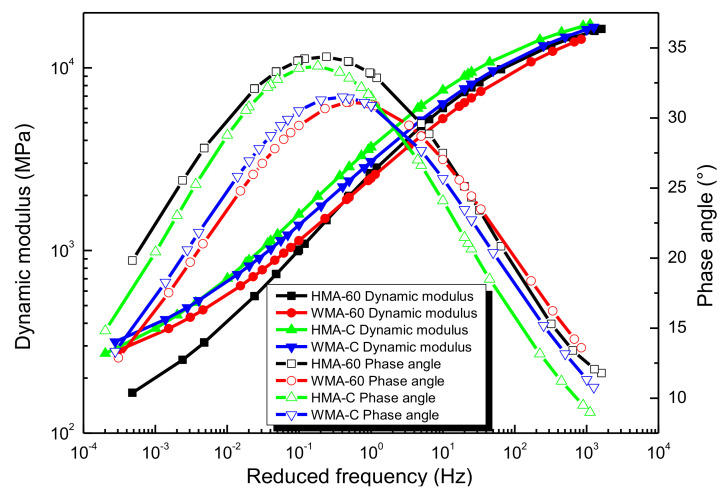
Dynamic modulus and phase angle of the asphalt mixtures [24].

**Figure 19 materials-13-05051-f019:**
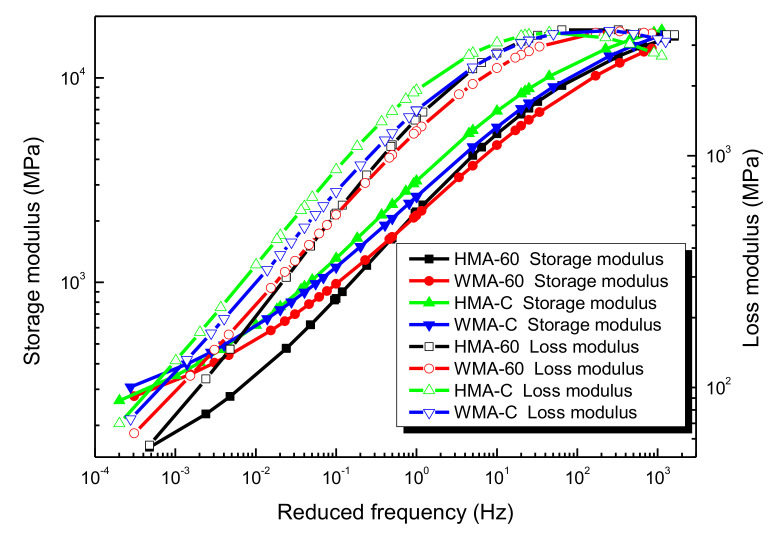
Storage modulus and loss modulus of the asphalt mixtures [24].

**Figure 20 materials-13-05051-f020:**
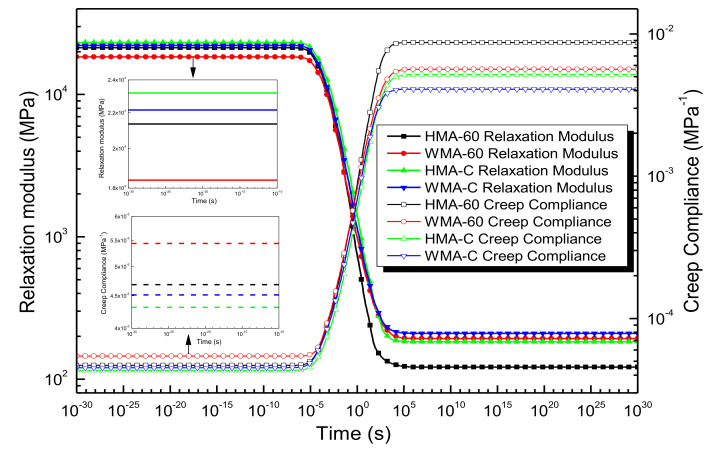
Relaxation modulus and creep compliance of the asphalt mixtures [24].

**Table 1 materials-13-05051-t001:** Aggregate blend percentages [24].

Aggregate	10–20 mm	5–10 mm	3–5 mm	0–3 mm	Filler
Blend Percentage by Weight/%	21	38	10	28	3

**Table 2 materials-13-05051-t002:** Volumetric parameters of crumb rubber modified asphalt mixtures at optimum asphalt content [24].

Asphalt Mixture	OAC/%	Gross Density g/cm³	Theoretical Density g/cm³	Void/%	VMA/%	VFA/%	Stability/KN	Flow Value/mm
HMA-60	5.4	2.439	2.537	3.86	14.04	72.5	10.45	2.74
WMA-60	5.4	2.442	2.538	3.78	13.93	72.8	10.95	2.87
HMA-C	5.6	2.441	2.546	4.12	14.15	70.9	11.23	2.45
WMA-C	5.6	2.444	2.546	4.01	14.04	71.5	11.61	2.62
Standard	-	-	-	3–5	≥ 13	65–75	≥ 8	2–4

**Table 3 materials-13-05051-t003:** Fitting parameters of the SM (Arrhenius equation).

Mixture Type	Parameters					Correlation	
	δ	α	β	γ	ΔE	$R_{E}^{2}\;$	$R_{\varphi }^{2}\;$
HMA-60	1.77	2.70	−0.55	−0.53	182899	0.998	0.911
WMA-60	2.18	2.30	−0.14	−0.29	158243	0.999	0.853
HMA-C	2.13	2.29	−0.53	−0.63	170479	0.998	0.976
WMA-C	2.18	2.34	−0.26	−0.55	174171	0.999	0.918

**Table 4 materials-13-05051-t004:** Fitting parameters of the SM (WLF equation).

Mixture Type	Parameters						Correlation	
	δ	α	β	γ	$C_{1}\;$	$C_{2}\;$	$R_{E}^{2}\;$	$R_{\varphi }^{2}\;$
HMA-60	1.83	2.55	−0.51	−0.67	9.68	95.12	0.998	0.928
WMA-60	2.19	2.24	−0.14	−0.62	21.02	221.02	0.999	0.901
HMA-C	2.15	2.24	−0.53	−0.68	21.94	214.27	0.998	0.978
WMA-C	2.24	2.19	−0.28	−0.64	15.40	150.75	0.999	0.951

**Table 5 materials-13-05051-t005:** Fitting parameters of the SM (Polynomial equation).

Mixture Type	Parameters						Correlation	
	δ	α	β	γ	a	b	$R_{E}^{2}\;$	$R_{\varphi }^{2}\;$
HMA-60	1.80	2.59	−0.53	−0.65	0.00092	−0.1061	0.998	0.945
WMA-60	2.19	2.24	−0.16	−0.62	0.00042	−0.0957	0.999	0.891
HMA-C	2.15	2.25	−0.53	−0.66	0.00044	−0.1033	0.998	0.980
WMA-C	2.23	2.19	−0.29	−0.64	0.00064	−0.1037	0.999	0.953

**Table 6 materials-13-05051-t006:** Fitting parameters of the GSM (Arrhenius equation).

Mixture Type	Parameters						Correlation	
	δ	α	β	γ	λ	ΔE	$R_{E}^{2}\;$	$R_{\varphi }^{2}\;$
HMA-60	1.70	2.86	−0.54	−0.47	0.80	182721	0.998	0.922
WMA-60	2.33	2.10	−0.10	−0.62	0.82	158255	0.999	0.876
HMA-C	2.10	2.44	−0.54	−0.50	0.55	170464	0.999	0.982
WMA-C	2.30	2.23	−0.31	−0.52	0.58	174172	0.999	0.923

**Table 7 materials-13-05051-t007:** Fitting parameters of the GSM (WLF equation) [24].

Mixture Type	Parameters							Correlation	
	δ	α	β	γ	λ	$C_{1}\;$	$C_{2}\;$	$R_{E}^{2}\;$	$R_{\varphi }^{2}\;$
HMA-60	1.62	2.84	−0.60	−0.56	0.80	8.93	87.75	0.999	0.935
WMA-60	2.29	2.13	−0.14	−0.62	0.81	19.14	202.14	0.999	0.884
HMA-C	2.12	2.40	−0.54	−0.51	0.51	19.13	187.95	0.999	0.978
WMA-C	2.33	2.14	−0.31	−0.56	0.52	14.02	138.11	0.999	0.955

**Table 8 materials-13-05051-t008:** Fitting parameters of the GSM (Polynomial equation).

Mixture Type	Parameters							Correlation	
	δ	α	β	γ	λ	a	b	$R_{E}^{2}\;$	$R_{\varphi }^{2}\;$
HMA-60	1.69	2.79	−0.57	−0.55	0.8	0.00094	−0.1060	0.999	0.948
WMA-60	2.29	2.14	−0.15	−0.61	0.75	0.00046	−0.0954	0.999	0.903
HMA-C	2.11	2.43	−0.55	−0.50	0.51	0.00050	−0.1029	0.999	0.981
WMA-C	2.32	2.17	−0.33	−0.55	0.50	0.00069	−0.1033	0.999	0.964

**Table 9 materials-13-05051-t009:** Discrete relaxation spectrum of crumb rubber-modified asphalt mixture [24].

*i*	HMA-60		WMA-60		HMA-C		WMA-C	
	*ρ_i_*	*E_i_*	*ρ_i_*	*E_i_*	*ρ_i_*	*E_i_*	*ρ_i_*	*E_i_*
1	2 × 10^−5^	3852.1	2 × 10^−5^	2650.0	2 × 10^−5^	3087.9	2 × 10^−5^	4910.2
2	2 × 10^−4^	5434.5	2 × 10^−4^	4580.1	2 × 10^−4^	5998.5	2 × 10^−4^	5693.3
3	2 × 10^−3^	6061.5	2 × 10^−3^	5233.2	2 × 10^−3^	5665.0	2 × 10^−3^	5464.6
4	2 × 10^−2^	3680.0	2 × 10^−2^	3208.0	2 × 10^−2^	4254.5	2 × 10^−2^	3717.2
5	2 × 10^−1^	1751.0	2 × 10^−1^	1476.5	2 × 10^−1^	2239.5	2 × 10^−1^	1920.1
6	2	566.5	2	610.9	2	1054.3	2	758.0
7	2 × 10^1^	223.2	2 × 10^1^	271.7	2 × 10^1^	394.9	2 × 10^1^	318.4
8	2 × 10^2^	35.8	2 × 10^2^	105.4	2 × 10^2^	165.9	2 × 10^2^	120.9
9	2 × 10^3^	11.8	2 × 10^3^	40.4	2 × 10^3^	30.0	2 × 10^3^	32.8
10	2 × 10^4^	5.3	2 × 10^4^	10.1	2 × 10^4^	4.1	2 × 10^4^	7.91
11	2 × 10^5^	1.9	2 × 10^5^	1.2	2 × 10^5^	1.5	2 × 10^5^	1.65
	*E_e_* = 121.58		*E_e_* = 193.10		*E_e_* = 183.34		*E_e_* = 209.75	

**Table 10 materials-13-05051-t010:** Discrete retardation spectrum of crumb rubber-modified asphalt mixture [24].

*i*	HMA-60		WMA-60		HMA-C		WMA-C	
	*τ_j_*	*D_j_*	*τ_j_*	*D_j_*	*τ_j_*	*D_j_*	*τ_j_*	*D_j_*
1	2.40 × 10^−5^	5.70 × 10^−5^	2.30 × 10^−5^	5.69 × 10^−7^	2.30 × 10^−5^	4.97 × 10^−6^	2.50 × 10^−5^	8.82 × 10^−6^
2	2.82 × 10^−4^	3.12 × 10^−5^	2.80 × 10^−4^	2.79 × 10^−5^	2.80 × 10^−4^	2.01 × 10^−5^	2.90 × 10^−4^	2.32 × 10^−5^
3	3.72 × 10^−3^	5.46 × 10^−5^	3.63 × 10^−3^	6.83 × 10^−5^	3.31 × 10^−3^	4.00 × 10^−5^	3.47 × 10^−3^	5.04 × 10^−5^
4	4.47 × 10^−2^	1.66 × 10^−4^	4.27 × 10^−2^	1.55 × 10^−4^	3.98 × 10^−2^	1.04 × 10^−4^	4.07 × 10^−2^	1.24 × 10^−4^
5	5.50 × 10^−1^	3.94 × 10^−4^	4.37 × 10^−1^	3.45 × 10^−4^	4.37 × 10^−1^	2.41 × 10^−4^	4.57 × 10^−1^	2.91 × 10^−4^
6	5.10	1.05 × 10^−4^	4.07	7.66 × 10^−4^	4.68	6.04 × 10^−4^	4.27	6.74 × 10^−4^
7	4.90 × 10^1^	2.01 × 10^−3^	3.63 × 10^1^	1.23 × 10^−3^	4.17 × 10^1^	1.32 × 10^−3^	3.80 × 10^1^	1.23 × 10^−3^
8	2.63 × 10^2^	3.00 × 10^−3^	2.95 × 10^2^	1.60 × 10^−3^	3.72 × 10^2^	1.43 × 10^−3^	3.09 × 10^2^	1.21 × 10^−3^
9	2.15 × 10^3^	1.66 × 10^−3^	2.40 × 10^3^	1.01 × 10^−3^	2.34 × 10^3^	1.01 × 10^−3^	2.34 × 10^3^	3.28 × 10^−4^
10	2.09 × 10^4^	2.68 × 10^−4^	2.09 × 10^4^	3.34 × 10^−3^	2.04 × 10^4^	2.94 × 10^−4^	2.09 × 10^4^	1.00 × 10^−4^
11	2.04 × 10^5^	1.70 × 10^−5^	2.04 × 10^5^	3.94 × 10^−5^	2.04 × 10^5^	5.94 × 10^−6^	2.04 × 10^5^	2.47 × 10^−6^
	*Dg* = 4.60 × 10^−5^		*Dg* = 5.4 × 10^−5^		*Dg* = 4.33 × 10^−5^		*Dg* = 4.41 × 10^−5^	

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
