# Peer review of "Predict the Phase Angle Master Curve and Study the Viscoelastic Properties of Warm Mix Crumb Rubber-Modified Asphalt Mixture"

_materials, 2020, doi:10.3390/ma13215051_

Round 1
Reviewer 1 Report
Introduction - Needs to be updated since it does not reflect the actual state of the art.
Figures 2 to 5 - Please introduce the error bars.
The authors state that:
"The relaxation modulus and creep compliance can be used to characterized the viscoelastic properties of the crumb rubber-modified asphalt mixture before and after warm mix additive was added."
This needs to be fully justified and supported.
Author Response
Response to Reviewer 1 Comments
Point 1: Introduction - Needs to be updated since it does not reflect the actual state of the art.
Response 1: Thank you for your remarkable question. In accordance with your advice, we have revised the Introduction, and the results shown as follow:
- Introduction
The mechanical behavior of viscoelastic materials is related to frequency and loading history [[1],[2]]. Asphalt material is a typical viscoelastic material, so viscoelastic behavior is essential for understanding road performance. Asphalt mixture will exhibit linear viscoelastic properties in small strain levels or limited number of cycles load [[3],[4],[5],[6]]. The complex modulus test in dynamic load is the most conventional method for studying the linear viscoelasticity of asphalt mixtures. It can obtain the dynamic modulus and phase angle of the asphalt mixture in the linear viscoelastic range. In general, it is easy to measure the dynamic modulus directly, but due to the limitations of experimental equipment, the measurement of the phase angle will inevitably be scattered, and even in some cases, it is difficult to obtain by using the experimental method. Therefore, it is particularly important to construct the phase angle of the asphalt mixture based on the results of dynamic modulus. Then it is vital to obtain the functional form of the dynamic modulus master curve. In the 1950s and 1960s, nomographs were used to characterize the rheological properties of asphalt binder and mixtures [[7]]. Besides the standard logistic sigmoidal equation [[8],[9]], The Weibulls equations [[10]], and the generalized logistic sigmoid equation [[11],[12]] were used to characterize the dynamic modulus master curve of asphalt binder and mixture. Meanwhile, Dickersen and Witt [[13]] proposed the relationship between phase angle, complex modulus, and frequency of asphalt binder. Christensen and Anderson [[14]] further simplified the above model and proposed the CA (Christensen-Anderson) model; Marasteanu and Anderson [[15]] proposed the CAM (Christensen-Anderson-Marasteanu) model based on the CA model. Unfortunately, the models proposed by Dickersen and Witt use independent parameters to construct the master curves of dynamic modulus and phase angle, which makes it difficult to satisfy the K-K (Kramers-Kronig) relations [[16]], and although other models ensure that the master curves of dynamic modulus and phase angle shared the same parameters, it is difficult to apply them to both asphalt binders and asphalt mixtures. To ensure that the master curve meets the K-K relations, Booij and Thoone [[17]] first proposed predicting the phase angle by the slope method of complex modulus versus frequency based on the generalized K-K relations. This approach was also used by Christensen and Anderson [14] in obtaining phase angle of the Christensen Anderson model. Rowe [[18]] proposed a similar form of S-shaped equation for the analysis of asphalt mixture. Based on the K-K relations, Mensching [[19]] used the SM (sigmoidal model) and GSM (generalized sigmoidal model) to construct the phase angle master curve and used the black space parameter to evaluate the low-temperature performance of the asphalt mixture. Oshone [[20]] estimated phase angles based on dynamic modulus data for asphalt mixtures and evaluated the validity of the predictions by black space diagram. Liu [[21]] also used the method to construct master curve model for the complex modulus of asphalt mixtures, which is highly accurate and consistent with linear viscoelastic theory. Nobakht [[22]] investigated the effects of aging on dynamic modulus and phase angle by conducting complex modulus tests on asphalt mixtures aged at nine different levels in laboratory, and the results showed that the phase angle prediction model based on the K-K relations could accurately predict the phase angle master curves after different aging. According to the principle of viscoelasticity, all of these research results show that the feasible to predict the master curve of phase angle from the dynamic modulus. The purpose of this work is to evaluate the effectiveness of using different shifting techniques to construct the master curve model of dynamic modulus and phase angle for crumb rubber-modified asphalt mixtures before and after warm mixing by using the TTSP (time-temperature superposition principle) [[23]]. In addition, the phase angle master curve can be predicted from the dynamic modulus master curve based on the K-K relations. And then, the master curve of storage modulus and loss modulus was obtained. Finally, the relaxation and creep properties of hot mix crumb rubber-modified asphalt mixtures (HMA) and warm mix crumb rubber-modified asphalt mixtures (WMA) were investigated by the relaxation modulus and creep compliance.
Point 2: Figures 2 to 5 - Please introduce the error bars.
Response 2: Thank you for your remarkable question. In accordance with your advice, we have introduced the error bars in figures 2 to 5. However, because the error is small, it is not significant in Figures 2-5, and for this reason the error values are listed in Table of the figures. The results are shown as following.
Figure 2 Shift factors of HMA-60.
Figure 3 Shift factors of WMA-60.
Figure 4 Shift factors of HMA-C.
Figure 5 Shift factors of WMA-C.
Point 3: The authors state that:
"The relaxation modulus and creep compliance can be used to characterized the viscoelastic properties of the crumb rubber-modified asphalt mixture before and after warm mix additive was added."
This needs to be fully justified and supported.
Response 3: Thank you for your remarkable question. This sentence is fully justified and supported by the following paragraph.
The GSM model is used to construct the dynamic modulus master curve, and the corresponding phase angle master curve is constructed based on the K-K relations. Both the Black space diagram and the Wicket diagram are used to verify that the constructed master curves compliance with the K-K relations, and then the master curves of storage modulus and loss modulus can be obtained with high accuracy. Based on the results of the storage modulus and loss modulus, relaxation modulus and creep compliance master curves can be obtained to investigate the relaxation and creep characteristics of the warm mix crumb rubber modified asphalt mixture.
In accordance with your advice, we have revised the sentence and shown it as following
" the relaxation modulus and creep compliance can be used to characterized the relaxation and creep properties of the warm mix crumb rubber-modified asphalt mixture "
References
[1] Schapery R A. Nonlinear viscoelastic solids. Int. J. Solids Struct., 2000, 37(1-2): 359-366, doi:10.1016/S0020-7683(99)00099-2.
[2] Hajikarimi P.; Fakhari Tehrani F.; Moghadas Nejad F.; et al. Mechanical behavior of polymer-modified bituminous mastics. I: experimental approach. J. Mater. Civ. Eng., 2019, 31(1): 04018337, doi:10.1061/(ASCE)MT.1943-5533.0002548.
[3] Lee H J.; Kim Y R. Viscoelastic constitutive model for asphalt concrete under cyclic loading. J. Eng. Mech., 1998, 124(1): 32-40, doi:10.1061/(ASCE)0733-9399(1998)124:11(1224).
[4] Gibson N H.; Schwartz C W.; Schapery R A.; et al. Viscoelastic, viscoplastic, and damage modeling of asphalt concrete in unconfined compression. Transp. Res. Record, 2003, 1860(1): 3-15, doi:10.3141/1860-01
[5] Nguyen Q T.; Di Benedetto H.; Sauzéat C. Linear and nonlinear viscoelastic behaviour of bituminous mixtures. Mater. Struct., 2015, 48(7): 2339-2351, doi:10.1617/s11527-014-0316-5.
[6] Diab A.; You Z.; Adhikari S.; et al. Modeling shear stress response of bituminous materials under small and large strains. Constr. Build. Mater., 2020, 252: 119133, doi:10.1016/j.conbuildmat.2020.119133.
[7] Heukelom W.; Klomp A J. Road design and dynamic loading. Assoc Asphalt Paving Technol Proc. 1964, 33, 92–125.
[8] Pellinen TK.; Witczak MW.; Bonaquist RF. Asphalt mix master curve construction using sigmoidal fitting function with non-linear least squares optimization technique. In: Proceedings of 15th ASCE engineering mechanics conference, Columbia University, New York, NY, June 2–5, 2002.
[9] Sirin O.; Paul D K.; Khan M S.; et al. Effect of Aging on Viscoelastic Properties of Asphalt Mixtures. J. Transp. Eng. B Pave, 2019, 145(4): 04019034, doi:10.1061/JPEODX.0000137.
[10] Weibull, W.; A statistical distribution function of wide applicability, J Appl Mech. 1951, 18: 290-293.
[11] Rowe, G. Phase Angle Determination and Interrelationships within Bituminous Materials 7th International RILEM Symposium on Advanced Testing and Characterization of Bituminous Materials; Taylor & Francis Group: London, UK, 2009; volume 1, pp. 43–52, ISBN: 9780415558563.
[12] Tanakizadeh A.; Shafabakhsh G. Viscoelastic characterization of aged asphalt mastics using typical performance grading tests and rheological-micromechanical models. Constr. Build. Mater., 2018, 188: 88-100, doi:10.1016/j.conbuildmat.2018.08.043.
[13] Dickinson E J.; Witt H P. The dynamic shear modulus of paving asphalts as a function of frequency. J. Rheol., 1974, 18(4): 591-606, doi:10.1617/s11527-016-0950-1.
[14] Christensen D W, Anderson D A. Interpretation of dynamic mechanical test data for paving grade asphalt cements (with discussion). Journal of Association of Asphalt Paving Technologists, 1992, 61, 67–116.
[15] Marasteanu M O.; Anderson D A. Improved model for bitumen rheological characterization, Eurobitume workshop on performance related properties for bituminous binders. Brussels, Belgium: European Bitumen Association, 1999, 133.
[16] Tschoegl, N.W. The Phenomenological Theory of Linear Viscoelastic Behavior: An Introduction; Springer: New York, NY, USA, 1989; ISBN: 9783642736025.
[17] Booij H C.; Thoone G. Generalization of Kramers-Kronig transforms and some approximations of relations between viscoelastic quantities. Rheol. Acta, 1982, 21(1): 15-24, doi:10.1007/BF01520701.
[18] Rowe G M.; Khoee S H.; Blankenship P.; et al. Evaluation of aspects of E* test by using hot-mix asphalt specimens with varying void contents. Transp. Res. Record, 2009, 2127(1): 164-172, doi:10.3141/2127-19.
[19] Mensching D J.; Rowe G M.; Sias Daniel J. A mixture-based Black Space parameter for low-temperature performance of hot mix asphalt. Road Mater. Pavement Des., 2017, 18(sup1): 404-425, doi:10.1080/14680629.2016.1266770.
[20] Oshone M.; Dave E.; Daniel J S.; et al. Prediction of phase angles from dynamic modulus data and implications for cracking performance evaluation. Road Mater. Pavement Des., 2017, 18(sup4): 491-513, doi:10.1080/14680629.2017.1389086.
[21] Liu H.; Luo R. Development of master curve models complying with linear viscoelastic theory for complex moduli of asphalt mixtures with improved accuracy. Constr. Build. Mater., 2017, 152: 259-268, doi: 10.1016/j.conbuildmat.2017.06.143.
[22] Nobakht M.; Sakhaeifar M S. Dynamic modulus and phase angle prediction of laboratory aged asphalt mixtures. Constr. Build. Mater., 2018, 190: 740-751, doi:10.1016/j.conbuildmat.2018.09.160.
[23] Nguyen Q T.; Di Benedetto H.; Sauzéat C.; et al. Time temperature superposition principle validation for bituminous mixes in the linear and nonlinear domains. J. Mater. Civ. Eng., 2013, 25(9): 1181-1188, doi:10.1061/(ASCE)MT.1943-5533.0000658.

Reviewer 2 Report
The paper is well written and interesting for the asphalt community.
Below some comments for minor revision of the manuscript:
1) Section 2.1 _ Please, specify the reason why 1% of warm mix additive was used.
2) Section 2.1 _ Please, specify the reason why 5.4 and 5.6% asphalt contents were used for mixes
3) Section 3.4 _ in the title, use "Determination of" instead of "Determine"
4) Section 4.4 _ in the title, use "Verification of" instead of "Verify"
5) Section 4.5, 4.5.1, 4.5.2, 4.5.3 _ in the title, "Compared" is not appropriate. I may suggest "Comparing"
Author Response
Point 1: The paper is well written and interesting for the asphalt community.
Response 1: Thank you for the compliment and we will definitely work harder in the future.
Point 2: Section 2.1 _ Please, specify the reason why 1% of warm mix additive was used.
Response 2: Thank you for your remarkable question. The content of warm mix additive is 1% by the weight of asphalt binder. The warm mix additive (SDYK) used in this paper is a surfactant. Theoretically, the addition of the surfactant can reduce the contact angle of the interface between asphalt and aggregate, so that the asphalt binder can be better spread on the aggregate during the mixing process, and the asphalt that penetrates into the matrix cannot enter the gaps, which helps to improve the encapsulation of the asphalt mixture. As one end of the surfactant molecules is hydrophilic, the other end is lipophilic, the lipophilic end of asphalt aggregates to form micelles, after mixing asphalt and aggregate, the surfactant molecules migrate to the interface between asphalt and aggregate, the lipophilic end combines with asphalt to form a lubricating layer. In the compaction process, due to the shear compaction of the roller and lipophilic base layer lubrication at the interface between the aggregate and asphalt, the construction of the asphalt mixture and the convenience of compaction is improved, making the pavement easy to compaction. When the content of warm mix additive is too small, it is difficult to play the role of lowering the construction temperature, and too much will increase the cost, manufacturers recommend the best content is 1% by the weight of asphalt binder.
Point 3: Section 2.1 _ Please, specify the reason why 5.4 and 5.6% asphalt contents were used for mixes
Response 3: Thank you for your remarkable question. The volumetric parameters of 60-mesh and compound-mesh crumb rubber modified asphalt mixes at optimum asphalt content are shown in Table 1. The asphalt binder content of HMA-C (5.6%) is greater than that of HMA-60 (5.4%), This is because the viscosity of the compound mesh crumb rubber-modified asphalt is greater than 60 mesh crumb rubber-modified asphalt, so more binder is required to coat the aggregates.
Table 1 Volume parameters of optimum asphalt content
|
Asphalt Mixture |
OAC/% |
Gross Density g/cm³ |
Theoretical Density g/cm³ |
Void/% |
VMA/% |
VFA/% |
Stability/KN |
Flow Value/mm |
|
HMA-60 |
5.4 |
2.439 |
2.537 |
3.86 |
14.04 |
72.5 |
10.45 |
2.74 |
|
WMA-60 |
5.4 |
2.442 |
2.538 |
3.78 |
13.93 |
72.8 |
10.95 |
2.87 |
|
HMA-C |
5.6 |
2.441 |
2.546 |
4.12 |
14.15 |
70.9 |
11.23 |
2.45 |
|
WMA-C |
5.6 |
2.444 |
2.546 |
4.01 |
14.04 |
71.5 |
11.61 |
2.62 |
|
Standard |
- |
- |
- |
3~5 |
≧13 |
65~75 |
≧8 |
2~4 |
As can be seen from Table 1, using the Marshall volume design method for grading design, the crumb rubber-modified bitumen content of 60 mesh crumb rubber-modified bitumen mixture (HMA-60) and composite mesh crumb rubber-modified bitumen mixture (HMA-C) is 5.4% and 5.6% by weight of the total mass of the mixture, respectively. This is because the viscosity of the compound mesh crumb rubber-modified asphalt is greater than 60 mesh crumb rubber-modified asphalt, so more binder is required to coat the aggregates. The sieving results of crumb rubber are presented in Table 2. In addition, the viscosity of the crumb rubber modified asphalt binder was presented in Table 3.
Table 2 The gradation of crumb rubber
|
Sieve No.(um) |
60 mesh crumb rubber |
compounded mesh crumb rubber |
||
|
Retained (%) |
Cumulative retained (%) |
Retained (%) |
Cumulative retained (%) |
|
|
40 (425) |
0 |
0 |
37.8 |
37.8 |
|
60 (250) |
9.2 |
9.2 |
40.1 |
77.9 |
|
80 (180) |
32.5 |
41.7 |
13.0 |
90.9 |
|
100 (150) |
33.7 |
75.4 |
7.3 |
98.2 |
|
120 (125) |
10.6 |
86.0 |
1.3 |
99.5 |
|
>120 (>75) |
14 |
100 |
0.5 |
100 |
Table 3 The Penetration Ductility and Softening point of crumb rubber
|
|
Penetration / 0.1mm |
Ductility / cm |
Softening point / ℃ |
Viscosity / Pa.s |
|
H-CR-60 |
73.2 |
16.5 |
53.9 |
1.45 |
|
W-CR-60 |
70.8 |
15.2 |
56.8 |
1.13 |
|
H-CR-C |
67.3 |
25.3 |
60.1 |
1.63 |
|
W-CR-C |
65.7 |
22.9 |
62.3 |
1.28 |
Point 4: Section 3.4 _ in the title, use "Determination of" instead of "Determine"
Response 4: Thank you for your remarkable question. In accordance with your advice, we have use "Determination of" instead of "Determine" in the section 3.4 _ in the title.
Point 5: Section 4.4 _ in the title, use "Verification of" instead of "Verify"
Response 5: Thank you for your remarkable question. In accordance with your advice, we have use "Verification of" instead of "Verify" in the section 4.4 _ in the title.
Point 6: Section 4.5, 4.5.1, 4.5.2, 4.5.3 _ in the title, "Compared" is not appropriate. I may suggest "Comparing" in the section 4.4 _ in the title.
Response 6: Thank you for your remarkable question. In accordance with your advice, we have use " Comparing " instead of " Compared " in the section 4.5, 4.5.1, 4.5.2, 4.5.3 _ in the title.

Reviewer 3 Report
This manuscript compares different master curve models for viscoelastic functions of selected crumb rubber modified-asphalt mixtures. The topic presents evident applicative interest, the experimental work, is very well conceived and conducted, data interpretation is reliable and referencing is appropriate.
However, I have a major concern on this paper, regarding the selected shift factors. In my opinion, from a rheological point of view, unique shift factors (at every temperature) must be used to build a unique master curve. It is not correct to use 3 types of shift factors depending on the model used. In fact, once the best factors are chosen, their dependence with temperature can be fit to the proposed models,
Some other detailed are shown below:
- Details of crumb rubber processing method must be provided: mixing temperature, processing time and temperature, mixing device etc.
- The paper contains too many figures, and most of them can be merged into one.
- I order to understand the role of the mineral aggregates for every mixture, master curves of the binder should be reported
In conclusion, I would like to recommend this manuscript for publishing after major revision.
Author Response
Point 1: This manuscript compares different master curve models for viscoelastic functions of selected crumb rubber modified-asphalt mixtures. The topic presents evident applicative interest, the experimental work, is very well conceived and conducted, data interpretation is reliable and referencing is appropriate.
Response 1: Thank you for the compliment and we will definitely work harder in the future.
Point 2: However, I have a major concern on this paper, regarding the selected shift factors. In my opinion, from a rheological point of view, unique shift factors (at every temperature) must be used to build a unique master curve. It is not correct to use 3 types of shift factors depending on the model used. In fact, once the best factors are chosen, their dependence with temperature can be fit to the proposed models.
Response 2: Thank you for your remarkable question. In fact, the shift factor is unique from a rheological point of view, but we don't know its exact value., so we have to obtain it by experimental results and mathematical methods according to viscoelastic theory. Therefore, using different models to calculate the shift factor will inevitably lead to errors. From Table 1, it can be seen that the parameters of the same type of shift factor constructed based on the SM model and the GSM model are similar, in a word, the shift factors is almost same, and although there are large differences in the parameters of different types of shift factors, there is no significant difference in the results of the shift factor calculation. Only the momentum modulus and phase angle measured at high temperature (low frequency) have a slight variation. In fact, in order to ensure that the values of the shift factors are identical for the different functional forms, it is possible to simultaneously fit all the master curves of different models and shift factors. the parameter results of shift factor were shown in table1. In addition, the results of shift factor were shown in figures 1 to 4. From Figures 1-4, it can be learned that there is no significant difference between the different forms of shift factors.
Table 1 the parameter results of Shift factor
|
|
SM |
GSM |
||||||||
|
Arrhenius |
WLF |
Polynomial |
Arrhenius |
WLF |
Polynomial |
|||||
|
ΔE |
C1 |
C2 |
a |
b |
ΔE |
C1 |
C2 |
a |
b |
|
|
HMA-60 |
182899 |
9.68 |
95.12 |
0.00092 |
-0.1061 |
182721 |
8.93 |
87.75 |
0.00094 |
-0.1060 |
|
WMA-60 |
158243 |
21.02 |
221.02 |
0.00042 |
-0.0957 |
158255 |
19.14 |
202.14 |
0.00046 |
-0.0954 |
|
HMA-C |
170479 |
21.94 |
214.27 |
0.00044 |
-0.1033 |
170464 |
19.13 |
187.95 |
0.00050 |
-0.1029 |
|
WMA-C |
174171 |
15.40 |
150.75 |
0.00064 |
-0.1037 |
174172 |
14.02 |
138.11 |
0.00069 |
-0.1033 |
Figure 1 Shift factors of HMA-60.
Figure 2 Shift factors of WMA-60.
Figure 3 Shift factors of HMA-C.
Figure 4 Shift factors of WMA-C.
Point 3: Details of crumb rubber processing method must be provided: mixing temperature, processing time and temperature, mixing device etc.
Response 3: Thank you for your remarkable question. The crumb rubber modified asphalt binder mixing used in this study was the wet process, in which the crumb rubber is added to the virgin asphalt binder (penetration grade 80/100) before introducing it in the asphalt mixture. The crumb rubber modified asphalt binder was produced in the laboratory at 180℃ for 30 min by an open blade mixer at a blending speed of 700 rpm [[1]]. The percentage of crumb rubber added for the crumb rubber modified asphalt binder was 20% by the weight of virgin asphalt. For the warm mix crumb rubber modified asphalt binder, the warm mix additive was added to crumb rubber modified asphalt binder mixing at 180 °C for 30 minutes by a conventional mechanical mixer.
Point 4: The paper contains too many figures, and most of them can be merged into one.
Response 4: Thank you for your remarkable question. We agree with you, but in fact, the differences of shift factors or dynamic modulus for mixtures in logarithmic coordinates are not significant, and it is difficult to reflect facts after merging the figures (For example, Figure 5). The authors would prefer not to merge figures and pray for discretion.
Figure 5 Shift factors of HMA-60 and WMA-60.
Point 5: I order to understand the role of the mineral aggregates for every mixture, master curves of the binder should be reported.
Response 5: Thank you for your remarkable question. In accordance with your advice, we have added master curves of the binder for every mixture. The master curves of complex modulus can be constructed by CAM model. Similarly, the master curve of phase angle can be obtained by shifting the measured phase angle horizontally in log frequency coordinates. The shift factor is consistent with the complex modulus. The results were shown in figure 6. The figure shows that the complex modulus master curve of the asphalt binder increases with the increase of frequency; conversely, the phase angle master curve decreases with the increase of frequency. The result is consistent with many literatures [[2],[3]].
Figure 6 master curve of crumb rubber modified asphalt binder.
In conclusion, I would like to recommend this manuscript for publishing after major revision.
References
[1] 1. Wang H, Li X, Xiao J, et al. High-Temperature Performance and Workability of Crumb Rubber–Modified Warm-Mix Asphalt. Journal of Testing and Evaluation, 2020, 48(4).
[2] Zhao Y, Chen P, Cao D. Extension of Modified Havriliak-Negami Model to Characterize Linear Viscoelastic Properties of Asphalt Binders[J]. Journal of Materials in Civil Engineering, 2015:04015195.
[3] Li Q, Li G, Ma X, et al. Linear viscoelastic properties of warm-mix recycled asphalt binder, mastic, and fine aggregate matrix under different aging levels[J]. Construction and Building Materials, 2018, 192(DEC.20):99-109.

Reviewer 4 Report
GENERAL COMMENTS
The paper presents an experimental and theoretical study aimed at: i) identifying the most accurate approach for the construction of the complex modulus master curves for warm asphalt rubber mixtures; ii) assessing the feasibility of predicting the phase angle master curves from the complex modulus ones.
Although the topic could be of interest for the field enhancing the existing literature, the paper is characterized by several critical flaws which negatively affect the overall quality of the manuscript.
The use of English is poor.
The abstract is confused; it does not clearly summarize the motivation, the methodologies and the main findings achieved.
The objective of the paper is not clearly defined, and it does not report any reference to the prediction of the phase angle master curves (lines 59-65).
Materials and methodologies are not well described; a colleague cannot reproduce the experiments with the information provided (e.g. what about preparation and storage of AR binder? How was the warm additive added? how did you chose the mixing and compaction temperatures? How do you identify the LVE limit? Etc.)
Only results of AR mixtures (two hot and two warm mixes) are presented. Results of the corresponding mixtures prepared with a plain binder would have been significant to really show the peculiarities of the investigated rubber modified materials.
Questionable or no justification/interpretation of the experimental observations are sometimes provided.
Section 4.3 includes a lot of repetitions of previous remarks as well as poor/basic/obvious comments.
Results reported in sections 4.5.2 and 4.5.3 seem out of the scope of the paper. Moreover, too many similarities with a work already published in Materials can be found in this part.
The conclusion part is affected by the abovementioned issue. Please be aware that the major scientific findings of the study should be properly summarized in the conclusions which should be concise and accurate.
Overall, the paper seems a pure mathematical exercise without a clear practical implication; the Authors should strengthen this aspect.
SPECIFIC COMMENTS
All the acronyms should be spelled out the first time they appear in the text (abstract excluded). E.g. CAM, K-K, GSM, etc.
Lines 27-28: a viscous material is time-dependent (time-dependent and frequency-dependent is the same thing); not necessarily thermo-dependent
Line 36: reference 10 is dated back 1958. You cannot use “then, …” if you cite an earlier reference
Line 45: [17] is less recent than [16] thus it cannot “solve the problem” emerged in [16]
Line 69: what do you mean with “fine”?
Line 69: Table 1 does not show the gradation
Line 72: what do you mean with “bias”?
Line 73: is SDYK a commercialism?
Lines 75-76: what do you mean with “60-mesh” and “compound-mesh”?
Line 82: is “produced by Australian IPC company” a commercialism?
Table 2 appears before than Table 1
Page 4 (and following): do not repeat the meaning of the symbols every time they appear
Line 132: not correct
Lines 133-135: actually, you measure stress and strain and then you can calculate E* and phase angle and, consequently, E’ and E’’
Page 4: not all the symbols of Eqs 4-6 are explained
Line 152: reference [34] does not seem appropriate
Line 154: it is not correct asserting that the minimum modulus value is nearly to zero
Line 92: is “produced by Australian IPC company” a commercialism?
Eq. 8: using apexes for the same variables as in Eq. 7 does not seem correct and could be misleading
Line 177: the word “model” is not appropriate since you are predicting the values based on E*, not modelling experimental findings
Tables 3-8: comparison of curves instead of parameters would have been more significant
Section 4.1: shift factors should be similar, otherwise shifted experimental values will not be correctly aligned. In fact, the discrepancy in the shift factors at the higher temperatures reflects in a master curve (e.g. HMA-60 Arrhenius) not consistent with the experimental findings (see figures 6, 7, etc.)
Section 4.2.1 (and following): showing only some results is questionable and suspicious. Why not all -60 mixtures? Or all the -C mixtures?
Very high R2 values reported in Tables 3-8 do not seem consistent with the discrepancies showed in the figures 6-10 depicting the master curves
Do not use “false” indicators in the figures. Use indicators only for experimental data and lines for models
Figure 11 (and following): the bell-shaped curves are not usual (normally the higher th reduced frequency the lower the phase angle). Please elaborate
Line 365: what is the Wicket diagram? Please explain in detail
Figure 17: what is the “loss factor”?
Line 375-379: obvious
Lines 393-405: no justification provided
Lines 417-420: since the phase angle is very low it is obvious that storage modulus is similar to dynamic modulus
Lines 424-427: obvious
Lines 428-429: not correct
From line 436: why are you talking about deformation resistance? Please explain
Lines 443-447: superficial and not clear if one does not know the difference between -60 and -C rubber
Lines 447-454: no experimental observations supporting these suppositions
Line 474: the format of this reference is not correct and this reference is not reported in the specific section at the end of the paper
Lines 486-491: and thus?
Lines 520-525: poor. Moreover, why are you talking about deformation resistance? Please explain
Author Response
GENERAL COMMENTS
The paper presents an experimental and theoretical study aimed at: i) identifying the most accurate approach for the construction of the complex modulus master curves for warm asphalt rubber mixtures; ii) assessing the feasibility of predicting the phase angle master curves from the complex modulus ones.
Although the topic could be of interest for the field enhancing the existing literature, the paper is characterized by several critical flaws which negatively affect the overall quality of the manuscript.
Point 1: The use of English is poor.
Response 1: Thank you for your remarkable question. In accordance with your advice, we have checked and carefully revised the English language and grammar of the entire paper.
Point 2: The abstract is confused; it does not clearly summarize the motivation, the methodologies and the main findings achieved.
Response 2: Thank you for your remarkable question. In accordance with your advice, we have revised the abstract and shown in here.
To identify the most accurate approach for the construction of the dynamic modulus master curves for warm mix crumb rubber modified asphalt mixtures, and assess the feasibility of predicting the phase angle master curves from the dynamic modulus ones. The SM (Sigmoidal model) and GSM (generalized sigmoidal model) were utilized to construct the master curve of dynamic modulus, respectively. The master curve of phase angle could be predicted from the master curve of dynamic modulus according to the K-K (Kramers-Kronig) relations. The results show that both SM and GSM can predict the dynamic modulus and phase angle well, except that the GSM shows slightly higher correlation coefficient than SM. Therefore, it is recommended to construct the dynamic modulus master curve using GSM and obtain the corresponding phase angle master curve based on the K-K relations. The Black space diagram and Wicket diagram were utilized to verify the predictions compliance with the linear viscoelastic theory. Then the master curve of storage modulus and loss modulus were also obtained. Finally, the relaxation modulus and creep compliance can be used to characterized the relaxation and creep properties of the warm mix crumb rubber-modified asphalt mixture
Point 3: The objective of the paper is not clearly defined, and it does not report any reference to the prediction of the phase angle master curves (lines 59-65).
Response 3: Thank you for your remarkable question. In accordance with your advice, we have revised the sentences in lines 59-65 and added the reference for it, the results shown in here.
The purpose of this work is to evaluate the effectiveness of using different shifting techniques to construct the master curve model of dynamic modulus and phase angle for crumb rubber-modified asphalt mixtures before and after warm mixing by using the TTSP (time-temperature superposition principle) [[1]]. In addition, the functional form of the phase angle master curve can be predicted from the dynamic modulus master curve based on the K-K relations [[2],[3]]. And then, the master curve of storage modulus and loss modulus was obtained. Finally, the relaxation and creep properties of hot mix crumb rubber-modified asphalt mixtures (HMA) and warm mix crumb rubber-modified asphalt mixtures (WMA) were investigated by the relaxation modulus and creep compliance.
Point 4: Materials and methodologies are not well described; a colleague cannot reproduce the experiments with the information provided (e.g. what about preparation and storage of AR binder? How was the warm additive added? how did you chose the mixing and compaction temperatures? How do you identify the LVE limit? Etc.).
Response 4: Thank you for your remarkable question.
Description of materials and methodologies:
The crumb rubber modified asphalt binder mixing used in this study was the wet process, in which the crumb rubber is added to the virgin asphalt binder (penetration grade 80/100) before introducing it in the asphalt mixture. The crumb rubber modified asphalt binder was produced in the laboratory at 180℃ for 30 min by an open blade mixer at a blending speed of 700 rpm [[4]]. The percentage of crumb rubber added for the crumb rubber modified asphalt binder was 20% by the weight of virgin asphalt. For the warm mix crumb rubber modified asphalt binder, the warm mix additive (the dosage was 1% by the weight of crumb rubber modified asphalt binder) was added to crumb rubber modified asphalt binder mixing at 180 °C for 30 minutes by a conventional mechanical mixer. Then the warm crumb rubber modified asphalt binder was prepared for manufacture specimen. The storage stability of binder.
Chose the mixing and compaction temperatures:
As a result of the investigations, the mixing and compaction temperature of the hot-mix binder-modified asphalt (AR) mixes were determined. These studies found that the production temperature for AR mixtures reaches 180 ℃[[5],[6]]. In addition, the mixing and compaction temperatures of the warm-mix binder-modified asphalt mixes could be determined based on the isovolumic principle. The mixing and compaction temperature results of the mixtures are presented in Table 1.
Table 1 mixing and compaction temperatures for types of mixture
|
Item |
Mixed Temperature |
Compacted Temperature |
Type of crumb rubber for binder |
Whether or not the warm mix additive was added |
|
HMA-60 |
180℃ |
170℃ |
60-mesh |
No |
|
WMA-60 |
162℃ |
152℃ |
60-mesh |
Yes |
|
HMA-C |
180℃ |
170℃ |
Compounded-mesh |
No |
|
WMA-C |
162℃ |
152℃ |
Compounded-mesh |
Yes |
Determining LVE limits:
The specimen shall be discarded at the end of any testing series at each temperature period, if the cumulative unrecovered permanent strain was found to be greater than 1500 microstrain, reduce the maximum loading stress level to half. Keep the test data up to this last resting period, discard the specimen, and use a new specimen for the rest of testing periods under reduced load conditions.
AASHTO Standard Test Method TP-15 recommended that a sample be used to assess the stress level required at any given temperature and frequency so that the resulting axial strain is between 75 to 125 microstrain [[7]]. In our study, the highest temperature is only 50°C, which is less than the highest temperature recommended by AASHTO 79-15 (75°C), so the test was specified with the maximum strain not exceeding 70με, which ensured both the test within the linear viscoelastic range and a higher test accuracy. the UTM automatically adjusted the applied load to limit the axial strain of the specimen within 70με.
Point 5: Only results of AR mixtures (two hot and two warm mixes) are presented. Results of the corresponding mixtures prepared with a plain binder would have been significant to really show the peculiarities of the investigated rubber modified materials.
Response 5: Thank you for your remarkable question. We agree with you, but considering the initial purpose of the test was to compare the linear viscoelasticity of the crumb rubber-modified asphalt mixture before and after the warm mix additive, we don’t prepared mixtures with a plain binder. In addition, we recognize the limitations of this approach, which does not show the peculiarities of the investigated rubber modified materials. And next step, we will conduct in detail the properties of asphalt mixtures prepared with plain binder.
Point 6: Questionable or no justification/interpretation of the experimental observations are sometimes provided.
Response 6: Thank you for your remarkable question. In accordance with your advice,
We have provided a reliable justification/interpretation for all experimental observations. Details of the changes can be found in the manuscript.
Point 7: Section 4.3 includes a lot of repetitions of previous remarks as well as poor/basic/obvious comments.
Response 7: Thank you for your remarkable question. In accordance with your advice, we have revised the repetitions remarks as well as poor/basic/obvious comments in Section 4.3, see manuscript P4-P6 for details.
Point 8: Results reported in sections 4.5.2 and 4.5.3 seem out of the scope of the paper. Moreover, too many similarities with a work already published in Materials can be found in this part.
Response 8: Thank you for your remarkable question. Results reported in sections 4.5.2 and 4.5.3 describe the relaxation modulus and creep compliance of crumb rubber-modified asphalt mixture and visualize the relaxation and creep properties of crumb rubber-modified asphalt mixture, which is an important addition to study of the viscoelastic properties of crumb rubber-modified asphalt mixture.
Point 9: The conclusion part is affected by the abovementioned issue. Please be aware that the major scientific findings of the study should be properly summarized in the conclusions which should be concise and accurate.
Response 9: Thank you for your remarkable question. In accordance with your advice,
We have summarized the major scientific findings of the study in the conclusion section and also simplified the conclusions as appropriate. The final results are shown in here.
This study involves constructing SM and GSM for dynamic modulus of crumb rubber-modified asphalt mixtures using different shifting techniques. The phase angle can be obtained from the dynamic modulus master curve by K-K relations. The Black space diagram and Wicket diagram were used to evaluate the predicted results of the dynamic modulus and phase angle. Finally, the relaxation modulus and creep compliance can be used to characterize the relaxation and creep properties of warm mix crumb rubber-modified asphalt mixture. Based on the results of this study, the following conclusions were drawn:
(1) The shift factor calculated by the Arrhenius equation is always smaller than the WLF equation and the second-order polynomial equation, and the higher temperature, the more significant.
(2) Both SM and GSM can be used as the master curve models of dynamic modulus, except that GSM presents slightly excellent fitting than SM.
(3) Compared with the laboratory results, the prediction of phase angles constructed based on the K-K relations shows a higher correlation coefficient. Moreover, The accuracy of the predicted phase angle depends on the accuracy of the dynamic modulus master curve.
(4) The Black space diagram and the Wicket diagram demonstrate that the master curve of dynamic modulus and phase angle is constructed by the slope method compliance linear viscoelastic theory.
(5) According to the viscoelastic theory, the storage modulus master curve and the loss modulus master curve can be obtained from the complex modulus test. Furthermore, the storage compliance master curve and the loss compliance master curve can also be obtained. Finally, the master curve of the relaxation modulus and creep compliance can be obtained in the region.
(6) From the results of dynamic modulus and phase angle, we can obtain that the deformation resistance of HMA-60 is not as good as HMA-C. Once the warm mix Additive was added, the mixture's deformation resistance in the low-frequency region (high temperature) will be improved, the viscous flow in the high-frequency region (low temperature) will also be enhanced. The WMA-C presents a better deformation resistance at high temperature, while WMA-60 presents better crack resistance at low temperature.
(7) From the results of relaxation modulus and creep compliance, we can obtain that HMA-60 presents a better deformation capacity than HMA-C at a shorter reduced time and a worse deformation resistance than HMA-C at a longer reduced time. Once the warm mix additive was added, the mixture’s deformation capacity was improved at a shorter reduced time. And improved the deformation resistance in longer reduced time.
SPECIFIC COMMENTS
Point 10: All the acronyms should be spelled out the first time they appear in the text (abstract excluded). E.g. CAM, K-K, GSM, etc.
Response 10: Thank you for your remarkable question. In accordance with your advice, we have checked and carefully revised this article and added the full name of all the acronyms the first time they appear.
For example: CA (Christensen-Anderson), CAM (Christensen-Anderson-Marasteanu), K-K (Kramers-Kronig), SM (Sigmoidal model), GSM (generalized Sigmoidal model), TTSP (Time-Temperature Superposition Principle), et al
Point 11: Lines 27-28: a viscous material is time-dependent (time-dependent and frequency-dependent is the same thing); not necessarily thermo-dependent.
Response 11: Thank you for your remarkable question. In accordance with your advice, we have use " The mechanical behavior of viscoelastic materials is related to frequency and loading history " instead of " The mechanical behavior of viscoelastic materials is related to temperature, frequency, loading history, and time"
Point 12: Line 36: reference 10 is dated back 1958. You cannot use “then, …” if you cite an earlier reference.
Response 12: Thank you for your remarkable question. In accordance with your advice, we have repositioned the references and removed the "then". the results shown in here.
In the 1950s and 1960s, nomographs were used to characterize the rheological properties of asphalt binder and mixtures [[8]]. Besides the standard logistic sigmoidal equation [[9],[10]], the Weibulls equations [[11]], and the generalized logistic sigmoid equation [[12],[13]] were used to characterize the dynamic modulus master curve of asphalt binder and mixture.
Point 13: Line 45: [17] is less recent than [16] thus it cannot “solve the problem” emerged in [16].
Response 13: Thank you for your remarkable question. In accordance with your advice, we have repositioned the references and removed the phrase " To solve this problem ". the results shown in here.
Meanwhile, Dickersen and Witt [[14]] proposed the relationship between phase angle, complex modulus, and frequency of asphalt binder. Christensen and Anderson [[15]] further simplified the above model and proposed the CA (Christensen-Anderson) model; Marasteanu and Anderson [[16]] proposed the CAM (Christensen-Anderson-Marasteanu) model based on the CA model. Unfortunately, the models proposed by Dickersen and Witt use independent parameters to construct the master curves of dynamic modulus and phase angle, which makes it difficult to satisfy the K-K (Kramers-Kronig) relations [[17]], and although other models ensure that the master curves of dynamic modulus and phase angle shared the same parameters, it is difficult to apply them to both asphalt binders and asphalt mixtures. To ensure that the master curve meets the K-K relations, Booij and Thoone [[18]] first proposed predicting the phase angle by the slope method of complex modulus versus frequency based on the generalized K-K relations.
Point 14: Line 69: what do you mean with “fine”?
Response 14: Thank you for your remarkable question. " Technical Specification for Construction of Highway Asphalt Pavement"[[19]] defined in the asphalt mixture of fine aggregate is natural sand, artificial sand (including mechanical sand) and rock crumbs with a particle size of less than 2.36 mm.
Point 15: Line 69: Table 1 does not show the gradation
Response 15: Thank you for your remarkable question. Table 1 shown different aggregate stockpiles blended by the percentages. We have used " Table 1 shown different aggregate stockpiles blended by the percentages, and Figure 1 show the gradation of the aggregates " instead of " Table 1 and Figure 1 show the gradation of the aggregates "
Point 16: Line 72: what do you mean with “bias”?
Response 16: Thank you for your remarkable question. Tires can be divided into radial and bias tires, and the rubber powder used in this paper was prepared from bias tires at ambient temperature.
Point 17: Line 73: is SDYK a commercialism?
Response 17: Thank you for your remarkable question. The warm mix additive (SDYK) is a kind of surfactant, it was purchased from Wuxi Dowrid Chemical Technology Co. The percentage of warm mix additive is 1% by weight of the crumb rubber modified bitumen binder, according to the manufacturer's recommendations.
Point 18: Lines 75-76: what do you mean with “60-mesh” and “compound-mesh”?
Response 18: Thank you for your remarkable question. The crumb rubber produced by mechanical shredding at ambient temperature was obtained from same source of bias tire. crumb rubber of 60-mesh and compound-mesh are used respectively, and the gradation and technical parameters are shown in Table 2, 3.
Table 2 The gradation of crumb rubber
|
Sieve No.(um) |
60 mesh crumb rubber |
compounded mesh crumb rubber |
||
|
Retained (%) |
Cumulative retained (%) |
Retained (%) |
Cumulative retained (%) |
|
|
40 (425) |
0 |
0 |
37.8 |
37.8 |
|
60 (250) |
9.2 |
9.2 |
40.1 |
77.9 |
|
80 (180) |
32.5 |
41.7 |
13.0 |
90.9 |
|
100 (150) |
33.7 |
75.4 |
7.3 |
98.2 |
|
120 (125) |
10.6 |
86.0 |
1.3 |
99.5 |
|
>120 (>75) |
14 |
100 |
0.5 |
100 |
Table 3 Technical parameters of crumb rubber
|
Item |
Unit |
Test result |
Methods |
||
|
60 mesh |
compound-mesh |
||||
|
Density |
g/cm3 |
0.38 |
0.38 |
GB/T 19208-2008 6.2.4 |
|
|
Ash |
% |
4.0 |
4.2 |
GB/T 4498-1997 |
|
|
Acetone extracts |
% |
12 |
13 |
GB/T 3516-2006 |
|
|
Rubber hydrocarbon content |
% |
63 |
65 |
GB/T 14837-1993 |
|
|
Fibre content |
% |
0 |
0 |
GB/T 19208-2008 6.2.3 |
|
|
Metal content |
% |
0.1 |
0.1 |
GB/T 19208-2008 6.2.2 |
|
In the production of crumb rubber modified asphalt, when the crumb rubber (60 mesh) blended with virgin asphalt, we call it 60-mesh crumb rubber modified asphalt binder; similarly, when the crumb rubber (compound mesh) blended with virgin asphalt, we call it compound -mesh crumb rubber modified asphalt binder.
Point 19: Line 82: is “produced by Australian IPC company” a commercialism?
Response 19: Thank you for your remarkable question. Superpave gyratory compaction is manufactured by IPC Australia, it is a commercial company. The company is an internationally recognised designer and manufacturer of high-quality material testing systems, established in Australia in 1981.
Point 20: Table 2 appears before than Table 1
Response 20: Thank you for your remarkable question. In accordance with your advice,
We have removed Table 1 before than Table 2.
Point 21: Page 4 (and following): do not repeat the meaning of the symbols every time they appear
Response 21: Thank you for your remarkable question. In accordance with your advice,
We have removed repeat the meaning of the symbols every time they appear.
Point 22: Line 132: not correct
Response 22: Thank you for your remarkable question. In accordance with your advice, we have revised the sentences (in line 132) and equations (4,5) as follow:
The dynamic modulus is the absolute value of the complex modulus, it can be calculated by the ratio of amplitude stress to amplitude strain, as shown in Equation (5).
(4)
(5)
Where: is the value of complex modulus, MPa; is the axial stress amplitude measured by load cell installed on the actuator, MPa; is the axial strain amplitude measured by LVDT; is the imaginary unit defined by ; is the value of storage modulus, MPa; is the value of loss modulus, MPa.
Point 23: Lines 133-135: actually, you measure stress and strain and then you can calculate E* and phase angle and, consequently, E’ and E’’
Response 23: Thank you for your remarkable question. In accordance with your advice, we have revised the sentences (in line 133-135) and equations (4, 5, 6) as follow:
The complex modulus can be estimated from the ratio of the stress input to the strain response, as shown in Equation (4). The dynamic modulus is the absolute value of the complex modulus, it can be calculated by the ratio of amplitude stress to amplitude strain, as shown in Equation (5). It reflects the strength characteristics of the asphalt mixture. The phase angle is time lagging of the last five loading cycles between stress and strain, it can be obtain following the Equation (6).
(4)
(5)
(6)
Where: is the value of complex modulus, MPa; is the axial stress amplitude measured by load cell installed on the actuator, MPa; is the axial strain amplitude measured by LVDT; is the imaginary unit defined by ; is the value of storage modulus, MPa; is the value of loss modulus, MPa; is the phase angle, °; is the average time lagging of the last five loading cycles between stress and strain, s; is the average time of the last five stress cycles, s.
Point 24: Page 4: not all the symbols of Eqs 4-6 are explained
Response 24: Thank you for your remarkable question. In accordance with your advice, we revised the equations (4, 5, 6) as follow:
(4)
(5)
(6)
Where: is the value of complex modulus, MPa; is the axial stress amplitude measured by load cell installed on the actuator, MPa; is the axial strain amplitude measured by LVDT; is the imaginary unit defined by ; is the value of storage modulus, MPa; is the value of loss modulus, MPa; is the phase angle, °; is the average time lagging of the last five loading cycles between stress and strain, s; is the average time of the last five stress cycles, s.
Point 25: Line 152: reference [34] does not seem appropriate
Response 25: Thank you for your remarkable question. In accordance with your advice, we have used the reference " Rowe, G.; Baumgardner G.; Sharrock M. Functional forms for master curve analysis of bituminous materials 7th International RILEM Symposium on Advanced Testing and Characterization of Bituminous Materials; Taylor & Francis Group: London, UK, 2009; volume 1, pp. 43–52, ISBN: 9780415558563. " instead of " Joseph P V.; Mathew G.; Joseph K.; et al. Dynamic mechanical properties of short sisal fibre reinforced polypropylene composites. Compos. Pt. A-Appl. Sci. Manuf., 2003, 34(3): 275-290, doi:10.1016/S1359-835X (02) 00020-9. "
Point 26: Line 154: it is not correct asserting that the minimum modulus value is nearly to zero
Response 26: Thank you for your remarkable question. In accordance with your advice, we have used " approach to a limiting equilibrium value as the loading frequency decreases to zero " instead of " approach to minimum modulus value (nearly zero) as the loading frequency decreases to zero "
Point 27: Line 92: is “produced by Australian IPC company” a commercialism?
Response 27: Thank you for your remarkable question. A closed-loop servo-hydraulic universal testing machine (UTM-100) is manufactured by IPC Australia, it is a commercial company. The company is an internationally recognised designer and manufacturer of high-quality material testing systems, established in Australia in 1981.
Point 28: Eq. 8: using apexes for the same variables as in Eq. 7 does not seem correct and could be misleading
Response 28: Thank you for your remarkable question. δ, α, β, and γ was used to represent the location and shape parameters of SM in Equation 7, and to avoid misleading, although , , , and could also be used for the location and shape parameters of GSM, the authors wish to leave them no change for the sake of easy differentiation between SM and GSM parameters.
Point 29: Line 177: the word “model” is not appropriate since you are predicting the values based on E*, not modelling experimental findings
Response 29: Thank you for your remarkable question. In accordance with your advice, we have removed the word "model" from headings 3.2 and 3.3.
Point 30: Tables 3-8: comparison of curves instead of parameters would have been more significant
Response 30: Thank you for your remarkable question and we agree with you. The results of the dynamic modulus and phase angle master curves have shown in Sections 4.2 and 4.3. Tables 3-8 only show the results of fitting parameter of the master curves.
Point 31: Section 4.1: shift factors should be similar, otherwise shifted experimental values will not be correctly aligned. In fact, the discrepancy in the shift factors at the higher temperatures reflects in a master curve (e.g. HMA-60 Arrhenius) not consistent with the experimental findings (see figures 6, 7, etc.)
Response 31: Thank you for your remarkable question. In fact, the shift factor is unique from a rheological point of view, but we don’t know the specific value, so we have to obtain it by experimental results and mathematical methods according to viscoelastic theory. Therefore, using different models to calculate the shift factor will inevitably lead to errors. From Table 4, it can be seen that the parameters of the same type of shift factor constructed based on the SM model and the GSM model are similar, in a word, the shift factors is almost same, and although there are large differences in the parameters of different types of shift factors, there is no significant difference in the results of the shift factor calculation. The dynamic modulus and phase angle measured at high temperature (low frequency) have a higher variation, making the shift factor also different. Of course, it is possible to fit all the master curves and their shift factors for the different models simultaneously in order to make the magnitude of the shift factors of the different methods consistent.
Table 4 the parameter results of Shift factor
|
|
SM |
GSM |
||||||||
|
Arrhenius |
WLF |
Polynomial |
Arrhenius |
WLF |
Polynomial |
|||||
|
ΔE |
C1 |
C2 |
a |
b |
ΔE |
C1 |
C2 |
a |
b |
|
|
HMA-60 |
182899 |
9.68 |
95.12 |
0.00092 |
-0.1061 |
182721 |
8.93 |
87.75 |
0.00094 |
-0.1060 |
|
WMA-60 |
158243 |
21.02 |
221.02 |
0.00042 |
-0.0957 |
158255 |
19.14 |
202.14 |
0.00046 |
-0.0954 |
|
HMA-C |
170479 |
21.94 |
214.27 |
0.00044 |
-0.1033 |
170464 |
19.13 |
187.95 |
0.00050 |
-0.1029 |
|
WMA-C |
174171 |
15.40 |
150.75 |
0.00064 |
-0.1037 |
174172 |
14.02 |
138.11 |
0.00069 |
-0.1033 |
The figure 1 (e.g. HMA-60) show no significant difference among the different shift factor techniques, especially for WLF equation and second-order polynomial equations. The specific manifestation is that the shift factor calculated by the Arrhenius equation is always smaller than the WLF equation and the second-order polynomial equation, and the higher temperature, the more significant.
Figure 1 Shift factors of HMA-60.
It is because the shift factors constructed based on the Arrhenius method are different from those constructed by other methods (WLF, quadratic polynomial) at higher temperatures (in the case of HMA-60), then the master curve constructed based on the Arrhenius method must be different from those constructed by other methods in the low frequency (high temperature) range, and also different from the measurements. Such a result is shown in Figure 2 and 3.
Figure 2 Sigmoidal dynamic modulus master curve of HMA-60.
Figure 3 Generalized sigmoidal dynamic modulus master curve of HMA-60.
Point 32: Section 4.2.1 (and following): showing only some results is questionable and suspicious. Why not all -60 mixtures? Or all the -C mixtures?
Response 32: Thank you for your remarkable question. In Section 4.2.1 (and following): Considering the page length limitations of the paper, only the results for HMA-60 and WMA-C are listed. In fact, all mixtures satisfy a similar pattern. For ease of comprehension, all results are supplemented in the Appendix A.
Point 33: Very high R2 values reported in Tables 3-8 do not seem consistent with the discrepancies showed in the figures 6-10 depicting the master curves
Response 33: Thank you for your remarkable question. The correlation coefficient is a measure of the linear correlation among variables.
Where r is correlation coefficient, is experimental results of sequence x; is the mean value of sequence x; is experimental results of sequence y; is the mean value of sequence y; n is the number of experimental.
The correlation coefficients of all the mixtures can be referred to the results of the Excel file “correlation coefficients” from which it can be seen that all the mixtures have high correlation coefficients, it is because there are only 28 sample points ,on the other hand the maximum order of magnitude of the measured data is greater than the minimum order of magnitude, which leads to the variability of the dynamic modulus of the fit data in the low frequency region.
Point 34: Do not use “false” indicators in the figures. Use indicators only for experimental data and lines for models
Response 34: Thank you for your remarkable question. It is customary to use dots to represent experimental data and lines to represent model curves, but sometimes we have to use dotted lines instead of lines because there are too many curves to distinguish between them.
Point 35: Figure 11 (and following): the bell-shaped curves are not usual (normally the higher the reduced frequency the lower the phase angle). Please elaborate
Response 35: For viscoelastic fluid of asphalts, the phase angle master curve is not usual bell-shaped. Generally, the higher the reduced frequency the lower the phase angle, as evidenced by many literatures [[20],[21],[22]]. For viscoelastic solid of asphalt mixtures, the phase angle curve is usually exhibit bell-shaped over a wide range of reduced frequency. Generally, increase with frequency, reaching a peak point, and then decreasing with further increases in frequency. When the frequency is close to zero or infinity, the phase angle is close to zero, and also there are many literatures [[23],[24],[25],[26]] demonstrating such results.
In this paper, the phase angle master curves of the asphalt binder and the mixture can be referred to figure 4 and figure 5. The results were shown as follows
Figure 4 the phase angle results of asphalt binder
Figure 5 the phase angle results of asphalt mixture
Point 36: Line 365: what is the Wicket diagram? Please explain in detail
Response 36: the wicket diagram is a log-log graph or semi-log graph of versus the loss factor ( ). The values of and were firstly calculated using the master curve models of and . The predicted values were then plotted against on a log-log graph or semi-log graph, as exhibited in Fig. 6. The wicket plot of each mixture type clearly illustrated that all data points were located on or very close to a unique smooth curve. This fact demonstrated that and were unique functions of each other, which confirmed the compliance of the constructed and master curves with the linear viscoelastic theory.
Figure 6 The wicket diagram of asphalt mixture for WMA-C.
Point 37: Figure 17: what is the “loss factor”?
Response 37: Thank you for your remarkable question. The loss factor ( ) is the ratio of the storage modulus to the loss modulus . The loss factor is usually used as the vertical coordinate in the Wicket diagram.
Point 38: Line 375-379: obvious
Response 38: Thank you for your remarkable question. In accordance with your advice, we have modified the sentence on lines 375-379 and shown as following:
It can be observed in Figure 18 that the master curve of the dynamic modulus exhibits the S-shaped, it means that the crumb rubber-modified asphalt mixtures mainly characterize elastic in higher frequency (or low temperature) but are viscous in lower frequency (or high temperature).
Point 39: Lines 393-405: no justification provided
Response 39: Thank you for your remarkable question. The justification provided as follow:
It is because the viscosity of the 60-mesh crumb rubber-modified asphalt binder is less than the compound-mesh crumb rubber-modified asphalt binder. The cohesion of the mixture produced from the former is less than the latter, which results in the 60-mesh crumb rubber-modified asphalt mixture with lower dynamic modulus and higher phase angle. Once warm mix additive was added, the mixing temperature of the mixture will be reduced. This process also reduces the aging of the asphalt binder. The fluidity of the asphalt binder was also improved, the aggregate can absorb more asphalt, and the content of the structural asphalt will increase, so the dynamic modulus of warm mix crumb rubber-modified asphalt mixture is greater than hot mix crumb rubber-modified asphalt mixture in lower frequency range (high temperature).
Point 40: Lines 417-420: since the phase angle is very low it is obvious that storage modulus is similar to dynamic modulus
Response 40: Thank you for your remarkable question. In accordance with your advice, we have modified the sentence on lines 417-420 and shown as following:
Figure 19 is the master curve of Storage modulus and Loss modulus for four kinds of asphalt mixture. It is obvious that the storage modulus master curve is similar to the dynamic modulus master curve, especially at high frequencies range. As the frequency increases, the storage modulus increases gradually, and there is the minimum value in the low-frequency region and the maximum value in the high-frequency region. The master curve is a typical S-shape. This is because the phase angle of the mixture is very low, especially at high frequencies, where it is approach to a very small magnitude.
Point 41: Lines 424-427: obvious
Response 41: Thank you for your remarkable question. Although this is obvious, to further explain the conclusions that followed, such results were presented here to improve the reliability of the relaxation and creep properties.
Point 42: Lines 428-429: not correct
Response 42: Thank you for your remarkable question. In accordance with your advice, we have used "Figure 19 illustrates that the master curve of the loss modulus. It can be seen from this figure that the loss modulus first increases and then slightly decreases as the frequency increases" instead of " Figure 19 also illustrated that the master curve of loss modulus is also similar to the master curve of the phase angle. As the frequency increases, the loss modulus increases and then decreases."
Point 43: From line 436: why are you talking about deformation resistance? Please explain
Response 43: Thank you for your remarkable question. In fact, the dynamic modulus can be used to describe the resistance to deformation of asphalt mixtures. The higher the dynamic modulus, the stronger the resistance to deformation; on the contrary, the poorer the resistance to deformation. The real part of the dynamic modulus (i.e., storage modulus) can be used to describe the asphalt mixture's resistance to elastic deformation, similarly, the imaginary part of the dynamic modulus (i.e., loss modulus) can also be used to describe the asphalt mixture's resistance to viscous deformation. In this paper, the changes in storage modulus and loss modulus were used to characterize the mixture's resistance to elastic deformation and viscous deformation, and also are used to indirectly study the high and low temperature properties of the mixtures.
Point 44: Lines 443-447: superficial and not clear if one does not know the difference between -60 and -C rubber
Response 44: Thank you for your remarkable question. As can be obtained from Table 5, the rotation viscosity of compound-mesh crumb rubber modified asphalt is greater than that of 60-mesh crumb rubber modified asphalt. The results was also shown in the appendix A.
Table 5 The Penetration Ductility and Softening point of crumb rubber
|
|
Penetration / 0.1mm |
Ductility / cm |
Softening point / ℃ |
Viscosity / cP |
|
H-CR-60 |
73.2 |
16.5 |
53.9 |
1452 |
|
W-CR-60 |
70.8 |
15.2 |
56.8 |
1148 |
|
H-CR-C |
67.3 |
25.3 |
60.1 |
1633 |
|
W-CR-C |
65.7 |
22.9 |
62.3 |
1384 |
Point 45: Lines 447-454: no experimental observations supporting these suppositions
Response 45: Thank you for your remarkable question. As can be obtained from Table 5, the rotation viscosity of compound mesh crumb rubber modified asphalt is greater than that of 60 mesh crumb rubber modified asphalt; once the warm mix additive was added, the viscosity of crumb rubber modified asphalt will be reduced. In addition, it can be obtained from Table 6 that the effective thickness of asphalt film of compound mesh crumb rubber modified asphalt is greater than that of 60 mesh crumb rubber modified asphalt; once the warm mix additive was added, the effective thickness of asphalt film of crumb rubber modified asphalt will be increased.
Table 6 Technical specifications for types of mixture
|
Item |
Mixed Temperature |
Compacted Temperature |
Content of crumb rubber modified asphalt binder |
Effective thickness of asphalt film |
|
HMA-60 |
180℃ |
170℃ |
5.4% |
9.2um |
|
WMA-60 |
162℃ |
152℃ |
5.4% |
9.5um |
|
HMA-C |
180℃ |
170℃ |
5.6% |
9.3um |
|
WMA-C |
162℃ |
152℃ |
5.6% |
9.7um |
Point 46: Line 474: the format of this reference is not correct and this reference is not reported in the specific section at the end of the paper
Response 46: Thank you for your remarkable question. In accordance with your advice, we have revised the format of reference on line 474, which is cross-referenced, and detailed reporting can be found In the reference section of the manuscript.
[41] Brinson H F.; Brinson L C. Polymer Engineering Science and Viscoelasticity: An Introduction. 2rd ed; Springer: New York, NY, USA, 2015; ISBN: 9781489974853.
Point 47: Lines 486-491: and thus?
Response 47: Thank you for your remarkable question. In accordance with your advice, we have added the following content:
This is also because the viscosity of the 60-mesh crumb rubber-modified bitumen binder is less than that of the compound-mesh crumb rubber-modified bitumen binder. Then the mixture produces by using the latter binder with more pronounced elastic properties, which resulting in the compound-crumb rubber-modified asphalt mixture with poor relaxation characteristics. Once the warm mix additive was added, the mix temperature of the mixture will be reduced. This process also reduced the aging of the asphalt mixture. In addition, it also improves the fluidity of the mixtures, so the warm-mix crumb rubber-modified asphalt mixture with better relaxation characteristics than the hot-mix crumb rubber-modified asphalt mixture in lower time range (low temperature).
Point 48: Lines 520-525: poor. Moreover, why are you talking about deformation resistance? Please explain
Response 48: Thank you for your remarkable question. In fact, the dynamic modulus can be used to describe the resistance to deformation of asphalt mixtures. The higher the dynamic modulus, the stronger the resistance to deformation; on the contrary, the poorer the resistance to deformation. The real part of the dynamic modulus (i.e., storage modulus) can be used to describe the asphalt mixture's resistance to elastic deformation, similarly, the imaginary part of the dynamic modulus (i.e., loss modulus) can also be used to describe the asphalt mixture's resistance to viscous deformation. In this paper, the changes in storage modulus and loss modulus were used to characterize the mixture's resistance to elastic deformation and viscous deformation, and also are used to indirectly study the high and low temperature properties of the mixtures.
References
[1] Nguyen Q T.; Di Benedetto H.; Sauzéat C.; et al. Time temperature superposition principle validation for bituminous mixes in the linear and nonlinear domains. J. Mater. Civ. Eng., 2013, 25(9): 1181-1188, doi:10.1061/(ASCE)MT.1943-5533.0000658.
[2] Booij H C.; Thoone G. Generalization of Kramers-Kronig transforms and some approximations of relations between viscoelastic quantities. Rheol. Acta, 1982, 21(1): 15-24, doi:10.1007/BF01520701.
[3] Oshone M.; Dave E.; Daniel J S.; et al. Prediction of phase angles from dynamic modulus data and implications for cracking performance evaluation. Road Mater. Pavement Des., 2017, 18(sup4): 491-513, doi:10.1080/14680629.2017.1389086.
[4] Wang H, Li X, Xiao J, et al. High-Temperature Performance and Workability of Crumb Rubber–Modified Warm-Mix Asphalt. Journal of Testing and Evaluation, 2020, 48(4).
[5] Gallego J, Castro M, Prieto JN, et al. Thermal sensitivity and fatigue life of gap-graded asphalt mixes incorporating crumb rubber from tire waste. Transport Res Rec J Transport Res Board, 2007.
[6] Rodríguez-Alloza, Ana María, Gallego J. Mechanical performance of asphalt rubber mixtures with warm mix asphalt additives. Materials & Structures, 2017, 50(2):147.
[7] AASHTO T P. Standard method of test for determining the dynamic modulus and flow number for asphalt mixture using the asphalt mixture performance tester. American Association of State Highway and Transportation Officials, Washington, DC, 2015.
[8] Heukelom W.; Klomp A J. Road design and dynamic loading. Assoc Asphalt Paving Technol Proc. 1964, 33, 92–125.
[9] Pellinen TK.; Witczak MW.; Bonaquist RF. Asphalt mix master curve construction using sigmoidal fitting function with non-linear least squares optimization technique. In: Proceedings of 15th ASCE engineering mechanics conference, Columbia University, New York, NY, June 2–5, 2002.
[10] Sirin O.; Paul D K.; Khan M S.; et al. Effect of Aging on Viscoelastic Properties of Asphalt Mixtures. J. Transp. Eng. B Pave, 2019, 145(4): 04019034, doi:10.1061/JPEODX.0000137.
[11] Weibull, W.; A statistical distribution function of wide applicability, J Appl Mech. 1951, 18: 290-293.
[12] Rowe, G. Phase Angle Determination and Interrelationships within Bituminous Materials 7th International RILEM Symposium on Advanced Testing and Characterization of Bituminous Materials; Taylor & Francis Group: London, UK, 2009; volume 1, pp. 43–52, ISBN: 9780415558563.
[13] Tanakizadeh A.; Shafabakhsh G. Viscoelastic characterization of aged asphalt mastics using typical performance grading tests and rheological-micromechanical models. Constr. Build. Mater., 2018, 188: 88-100, doi:10.1016/j.conbuildmat.2018.08.043.
[14] Dickinson E J.; Witt H P. The dynamic shear modulus of paving asphalts as a function of frequency. J. Rheol., 1974, 18(4): 591-606, doi:10.1617/s11527-016-0950-1.
[15] Christensen D W, Anderson D A. Interpretation of dynamic mechanical test data for paving grade asphalt cements (with discussion). Journal of Association of Asphalt Paving Technologists, 1992, 61, 67–116.
[16] Marasteanu M O.; Anderson D A. Improved model for bitumen rheological characterization, Eurobitume workshop on performance related properties for bituminous binders. Brussels, Belgium: European Bitumen Association, 1999, 133.
[17] Tschoegl, N.W. The Phenomenological Theory of Linear Viscoelastic Behavior: An Introduction; Springer: New York, NY, USA, 1989; ISBN: 9783642736025.
[18] Booij H C.; Thoone G. Generalization of Kramers-Kronig transforms and some approximations of relations between viscoelastic quantities. Rheol. Acta, 1982, 21(1): 15-24, doi:10.1007/BF01520701.
[19] Chinese National Specification JTG F40, 2004, “Technical Specification for Construction of Highway Asphalt Pavement,” People’s Communication Press, Beijing (in Chinese).
[20] Zhao Y, Chen P, Cao D. Extension of Modified Havriliak-Negami Model to Characterize Linear Viscoelastic Properties of Asphalt Binders[J]. Journal of Materials in Civil Engineering, 2015:04015195.
[21] Li Q, Li G, Ma X, et al. Linear viscoelastic properties of warm-mix recycled asphalt binder, mastic, and fine aggregate matrix under different aging levels[J]. Construction and Building Materials, 2018, 192(DEC.20):99-109.
[22] Wang D, Falchetto A C, Riccardi C, et al. Investigation on the low temperature properties of asphalt binder: Glass transition temperature and modulus shift factor[J]. Construction and Building Materials, 2020, 245:118351.
[23] Nobakht M, Sakhaeifar M S. Dynamic modulus and phase angle prediction of laboratory aged asphalt mixtures[J]. Construction and Building Materials, 2018, 190(NOV.30):740-751.
[24] Li, PeiLong, Zhan Ding, et al. Effect of Temperature and Frequency on Visco-Elastic Dynamic Response of Asphalt Mixture. Journal of Testing & Evaluation, 2013.
[25] Yang X, You Z. New Predictive Equations for Dynamic Modulus and Phase Angle Using a Nonlinear Least-Squares Regression Model[J]. Journal of Materials in Civil Engineering, 2014, 27(3):04014131.1-04014131.8.
[26] Liu, H., and Luo, R.* (2017). “Development of master curve models complying with linear viscoelastic theory for complex moduli of asphalt mixtures with improved accuracy.” Construction and Building Materials, Vol. 152, pp. 259–268.

Round 2
Reviewer 1 Report
All the issues raised were well addressed by the authors. Therefore, I support publication in the present form.
Reviewer 3 Report
The proposed suggestions/revisions have taken into consideration. Therefore in my opinion the manuscript is now acceptable for publication
Reviewer 4 Report
The Authors have tried to address all the points reported in the first revision.
However, several reviewer's remarks seem still valid even if they do not prevent the publication of the paper.